

# Global-scale combustion sources of organic aerosols: Sensitivity to formation and removal mechanisms

**Alexandra P. Tsimpidi[1], Vlassis A. Karydis[1], Spyros N. Pandis[2, 3]and Jos Lelieveld[1, 4]**

[1] Max Planck Institute for Chemistry, Mainz, Germany

[2] Department of Chemical Engineering, University of Patras, Patras, Greece

[3] Department of Chemical Engineering, Carnegie Mellon University, Pittsburgh, PA, USA

[4] Energy, Environment and Water Research Center, Cyprus Institute, Nicosia, Cyprus

*Corresponding author e-mail: a.tsimpidi@mpic.de



**Abstract**
Organic compounds from combustion sources such as biomass burning and fossil
fuel use are major contributors to the global atmospheric load of aerosols. We
analyzed the sensitivity of model-predicted global-scale organic aerosols (OA) to
parameters that control primary emissions, photochemical aging and the scavenging
efficiency of organic vapors. We used a computationally efficient module for the
description of OA composition and evolution in the atmosphere (ORACLE) of the
global chemistry-climate model EMAC. A global dataset of aerosol mass
spectrometer measurements was used to evaluate simulated primary (POA) and
secondary OA (SOA) concentrations. Model results are sensitive to the emission rates
of intermediate volatility organic compounds (IVOCs) and POA. Assuming enhanced
reactivity of semi-volatile organic compounds (SVOCs) and IVOCs with OH
substantially improved the model performance for SOA. Use of a hybrid approach for
the parameterization of the aging of IVOCs had a small effect on predicted SOA
levels. The model performance improved by assuming that freshly emitted organic
compounds are relatively hydrophobic and become increasingly hygroscopic due to
oxidation.
**1 Introduction**
Organic aerosol (OA) is an important constituent of the atmosphere, contributing
30-70% of the total submicron dry aerosol mass (Kanakidou et al., 2005) with major
impacts on human health and climate (IPCC, 2013; Lelieveld et al., 2015). OA
comprises a large number of compounds with a wide range in volatility and oxidation
states. The material that is in the particulate phase upon emission is called primary
organic aerosol (POA). The co-emitted organic vapors can undergo one or more
chemical transformations, which can reduce their volatility, leading to their transfer to
the particulate phase forming secondary organic aerosol (SOA).
Several regional-scale modeling studies have accounted for the semi-volatile
nature and chemical aging of organic compounds by using the volatility based set
(VBS) approach (Donahue et al., 2006), demonstrating improvements in the accuracy
of the predicted concentrations of organic aerosols (OA) and their chemical properties
(Robinson et al., 2007; Shrivastava et al., 2008; Murphy and Pandis, 2009; Hodzic et
al., 2010; Tsimpidi et al., 2010; Fountoukis et al., 2011; Li et al., 2011; Tsimpidi et





al., 2011; Bergstrom et al., 2012; Athanasopoulou et al., 2013; Zhang et al., 2013;
Fountoukis et al., 2014). However, only few global modeling studies have adopted the
VBS approach (Pye and Seinfeld, 2010; Jathar et al., 2011; Tsimpidi et al., 2014).
According to these studies, the modeled global tropospheric burden of POA is 0.03-
0.23 Tg and of SOA 1.61-2.77 Tg, with SVOCs and IVOCs contributing 0.71-1.57 Tg
to the total.
The VBS approach is a flexible framework for simulating OA formation and
removal; however, there are several uncertainties in the parameters used. The first
source of uncertainty is related to the emissions of organic particles and vapors
(Kanakidou et al., 2005). The volatility distribution of the fresh POA is important in
the VBS as it determines the initial evaporation of POA. Part of the IVOC emissions
is not included in conventional inventories, even if it is important for the predicted
SOA (Shrivastava et al., 2008; Grieshop et al., 2009; Tsimpidi et al., 2010). Several
studies have assumed a 50% addition to the traditional emission inventory (e.g.,
Shrivastava et al., 2008; Jathar et al., 2011; Tsimpidi et al., 2014) for IVOC emissions
but enhancements up to a factor 6.5 have been used in the literature (e.g., Shrivastava
et al., 2011). Furthermore, most previous modeling studies typically assumed the
same volatility distributions of all emissions independent of their source (e.g.
Robinson et al., 2007). However, recent investigations reported significant differences
in the volatility distribution of particles emitted from biomass burning, diesel and
gasoline vehicle exhausts (May et al., 2013a; May et al., 2013c, b).
The second source of uncertainty is related to the oxidation of the emitted SVOCs
and IVOCs. The parameters used by the VBS to simulate this process are the
oxidation rate constant, the volatility distribution of the products, and the oxygen
mass added per generation of oxidation. The VBS volatility resolution used to
represent the SVOC/IVOC volatility range ($3.2 \times 10^{-1}$ µg m$^{-3}$ $< C^* < 3.2 \times 10^{6}$ µg m$^{-3}$)
affects these parameters as well. A coarse volatility resolution requires a lower
effective oxidation rate constant and a more rapid addition of oxygen and reduction in
volatility than a finer volatility resolution. A common representation for the oxidation
of SVOCs and IVOCs, mainly used by regional models (e.g. Murphy and Pandis,
2009; Tsimpidi et al., 2010; Fountoukis et al., 2011; Tsimpidi et al., 2011; Bergstrom
et al., 2012; Athanasopoulou et al., 2013; Fountoukis et al., 2014), is based on the
work of Robinson et al. (2007) and Shrivastava et al. (2008) and includes 9 volatility





bins with saturation concentrations ranging from $10^{-2}$ to $10^{6}$ µg m$^{-3}$, an oxidation rate
constant of $4\times10^{-11}$ cm$^{3}$ molec$^{-1}$ s$^{-1}$ based on Atkinson and Arey (2003), a reduction in
volatility by one order of magnitude after each reaction, and a 7.5% net increase in
mass to account for the added oxygen. This formulation is rather conservative
compared to other studies which have assumed higher reduction in volatility and/or
increase in mass. Shrivastava et al. (2011) assumed a 15% increase in mass due to the
added oxygen, while Grieshop et al. (2009) and Hodzic et al. (2010) assumed a 40%
increase in mass and two orders of magnitude reduction in volatility in each reaction
step. Pye and Seinfeld (2010) simulated the POA emissions using two SVOCs (with
$C^{*}$ equal to 20 and 1646 µg m$^{-3}$) and one IVOC ($10^{5}$ µg m$^{-3}$) and used an oxidation
rate constant of $2\times10^{-11}$ cm$^{3}$ molec$^{-1}$ s$^{-1}$, two orders of magnitude reduction in
volatility in each reaction, and 50% increase in mass per reaction. Shrivastava et al.
(2011) used only two surrogate species ($C^{*}$ equal to $10^{-2}$ and $10^{5}$ µg m$^{-3}$), an oxidation
rate constant of $0.57\times10^{-11}$ cm$^{3}$ molec$^{-1}$ s$^{-1}$, seven orders of magnitude reduction in
volatility, and 50% increase in mass per reaction. Tsimpidi et al. (2014) used a lower
resolution VBS scheme with 4 surrogate species (with $C^{*}$ $10^{-1}$, $10^{1}$, $10^{3}$, and $10^{5}$ µg
m$^{-3}$), an oxidation rate constant of $2\times10^{-11}$ cm$^{3}$ molec$^{-1}$ s$^{-1}$, two orders of magnitude
reduction in volatility, and 15% increase in mass per reaction. All of the above
schemes should be viewed as parameterizations of the complex reactions that actually
take place; the oxidation products can be up to four orders of magnitude lower in
volatility than the precursor (Kroll and Seinfeld, 2008). To address this limitation,
Jathar et al. (2012) developed a hybrid method to represent the formation of SOA
from non-speciated SVOC and IVOC vapors. According to this framework, the first
generation of oxidation of SVOC and IVOC is parameterized by fitting to SOA data
from smog chamber experiments. Subsequently, the generic multi-generational
oxidation scheme of Robinson et al. (2007) was used for the subsequent generation
steps.

106        The third source of uncertainty is related to the scavenging efficiency of gas-phase

oxidized SVOCs and IVOCs. The water solubility of these organic vapors is largely
unknown and in most OA modeling studies a fixed effective Henry's law constant
(e.g., $H = 10^{5}$ M atm$^{-1}$) is used for all organic compounds. However, organic vapors
become increasingly more hydrophilic during their atmospheric lifetime. Pye and
Seinfeld (2010) treated the freshly emitted gas-phase SVOCs as relatively



hydrophobic ($H$= 9.5 M atm$^{-1}$) and their oxidation products as moderately hydrophilic
($H$= $10^5$ M atm$^{-1}$). Hodzic et al. (2014) argued that Henry's law constants have a
strong negative correlation with the saturation vapor pressures and depend on the
precursor species, the extent of photochemical processing, and the NO$_x$ levels during
the formation.
In this work we use ORACLE, a computationally efficient module for the
description of OA composition and evolution in the atmosphere (Tsimpidi et al.,
2014), to quantify the impact of the main VBS parameters on the model OA
predictions. Our main focus is the formation of OA from anthropogenic combustion
and open biomass burning sources. We conducted different tests to study the
sensitivity of the model predictions to emissions, photochemical aging and scavenging
efficiency of LVOCs, SVOCs and IVOCs. The results are compared to the reference
simulation and aerosol mass spectrometer (AMS) measurements at multiple locations
worldwide following Tsimpidi et al. (2016). Results from these sensitivity tests help
identify the major uncertainties of the VBS formulations and give rise to suggestions
about potential model improvements.

## 129    2 Reference model description and application

### 130    2.1 EMAC Model

The ECHAM/MESSy Atmospheric Chemistry (EMAC) model is a numerical
chemistry and climate simulation system that includes sub-models describing lower
and middle atmosphere processes and their interaction with oceans, land and human
influences (Jöckel et al., 2006). EMAC includes submodels that describe gas-phase
chemistry (MECCA; Sander et al., 2011), inorganic aerosol microphysics (GMXe;
Pringle et al., 2010), cloud microphysics (CLOUD; Jöckel et al., 2006), aerosol
optical properties (AEROPT; Lauer et al., 2007), dry deposition and sedimentation
(DRYDEP and SEDI; Kerkweg et al., 2006a), cloud scavenging (SCAV; Tost et al.,
2006), emissions (ONLEM and OFFLEM; Kerkweg et al., 2006b), and organic
aerosol formation and growth (ORACLE; Tsimpidi et al., 2014). The spectral
resolution used in this study is T63L31, corresponding to a horizontal grid spacing of
1.875$^o$x1.875$^o$ and 31 vertical layers extending to 25 km altitude. The 11-year period
between 2000 and 2010 is simulated, with the first year used as spin-up.



## 2.2 ORACLE Module


ORACLE is a computationally efficient submodel for the description of organic
aerosol composition and evolution in the atmosphere (Tsimpidi et al., 2014). ORACLE
simulates a wide variety of semi-volatile organic products separating them into bins of
logarithmically spaced effective saturation concentrations. In this study, primary
organic emissions from biomass burning and fuel combustion sources are taken into
account using separate surrogate species for each source category. These surrogates are
subdivided into three groups of organic compounds: LVOCs ($C^*=10^{-2}$ µg m$^{-3}$), SVOCs
($C^*=10^0$ and $10^2$ µg m$^{-3}$) and IVOCs ($C^*=10^4$ and $10^6$ µg m$^{-3}$). These organic
compounds are allowed to partition between the gas and aerosol phases resulting in the
formation of POA. Anthropogenic and biogenic VOCs are simulated separately, and
their oxidation results in products distributed in four volatility bins with effective
saturation concentrations $10^0$, $10^1$, $10^2$, and $10^3$ µg m$^{-3}$. Gas-phase photochemical
reactions that modify the volatility of the organics are taken into account and the
oxidation products (SOA-sv, SOA-iv, and SOA-v) of each group of precursors
(SVOCs, IVOCs, and VOCs) are simulated separately in the module to keep track of
their origin. LVOCs are not allowed to participate in photochemical reactions since
they are already in the lowest volatility bin. In total 52 organic compounds are
simulated explicitly (26 in each of the gas and aerosol phases). The volatilities of
SVOCs and IVOCs are reduced by a factor of $10^2$ as a result of the OH reaction with a
rate constant of $2\times10^{-11}$ cm$^3$ molecule$^{-1}$ s$^{-1}$ and a 15% increase in mass to account for
two added oxygens (Tsimpidi et al., 2014). The model set-up and the different aerosol
types and chemical processes that simulated by ORACLE in this study are illustrated
in Figure 1a. More details about ORACLE can be found in Tsimpidi et al. (2014).

## 2.3 Emission inventory


The CMIP5 RCP4.5 emission inventory (Clarke et al., 2007) is used for the
anthropogenic primary organic aerosol emissions from fuel combustion and biomass
burning. The open biomass burning emissions from savanna and forest fires are based
on the Global Fire Emissions Database (GFED v3.1; van der Werf et al., 2010). In
order to convert the emitted organic carbon (OC) to organic mass (OM), OM/OC
factors of 1.3 and 1.6 have been used for the anthropogenic and biomass burning
emissions, respectively (Aiken et al., 2008; Canagaratna et al., 2015). Furthermore,





emission fractions are used to distribute the OM to the volatility bins used by
ORACLE. The sum of the emission fractions used for the volatility bins with $C^* \leq$
$10^4$ is unity since current emission inventories are based on samples collected at
aerosol concentrations up to $10^4$ µg m$^{-3}$ (Shrivastava et al., 2008; Robinson et al.,
2010). Additional emission fractions can be assigned to the volatility bins with $C^* >$
$10^4$ based on dilution experiments (Robinson et al., 2007).
In this study we assume that anthropogenic fuel (fossil and biofuel) combustion
emissions cover a range of volatilities from $10^{-2}$ to $10^6$ µg m$^{-3}$ and the additional
IVOC emissions are 1.5 times the traditional POA emissions (Robinson et al., 2007);
therefore, the sum of the emission fractions for the fuel combustion emissions is 2.5
(Figure 2a). Biomass burning emissions are assumed to cover a range of volatilities
from $10^{-2}$ to $10^4$ (May et al., 2013a), therefore, no IVOC emissions are assumed from
biomass burning sources and the sum of their emission factors is unity (Figure 2a).
Overall, the decadal average global emission flux of primary organic emissions is 44
Tg yr$^{-1}$ from anthropogenic combustion sources and 28 Tg yr$^{-1}$ from open biomass
burning sources.

**3 Sensitivity Simulations**
All sensitivity calculations are conducted for the same 11-year period as the
reference simulation, the results of which have been analyzed by (Tsimpidi et al.,
2016). Table 1 summarizes the general characteristics of the sensitivity simulations. A
detailed description is provided below.

**3.1 Sensitivities to emissions**
The emissions of LVOCs, SVOCs and IVOCs are a key input for the accurate
description of atmospheric OA. To quantify the sensitivity of the reference case
results to the LVOC, SVOC and IVOC emissions, three simulation tests have been
designed. Figure 2 summarizes the emission factors used for the volatility distribution
of the emissions and the emission rate of each volatility bin for the reference and the
sensitivity tests. More specifically:

**Low volatility:** In this sensitivity simulation, we assume zero emissions of IVOCs to
quantify their contribution to the formation of global SOA. Therefore, the fuel



combustion and biomass burning emissions are distributed only in the LVOCs ($10^{-2}$
µg m$^{-3}$) and SVOCs ($10^0$ and $10^2$ µg m$^{-3}$) volatility bins and the sum of their emission
fractions is equal to unity (Figure 2b). The decadal average global emission flux of
primary organic emissions in this test is 18 Tg yr$^{-1}$ from anthropogenic combustion
sources and 28 Tg yr$^{-1}$ from open biomass burning sources (Table 1).

**High IVOCs:** To estimate an upper limit of the IVOC contribution to the formation
of SOA, a sensitivity simulation is conducted where the emissions of IVOCs are
increased by an additional factor of 1.5 times the POA emissions distributed in the
volatility bins with $C^*$ of $10^4$ and $10^6$ µg m$^{-3}$ (Figure 2c). The LVOC and SVOC
emissions are the same as in the reference simulation. Overall, the decadal average
global emission flux of primary organic emissions in this sensitivity test is 71 Tg yr$^{-1}$
for both anthropogenic and open biomass burning sources (Table 1).

**Alternative POA emissions:** To investigate the sensitivity of the model results to the
magnitude of the POA emissions, we have utilized the AEROCOM database for the
POA emissions from anthropogenic combustion sources (Dentener et al., 2006) and
the CMIP5 RCP4.5 emission inventory for the POA emissions from open biomass
burning sources. These emission inventories include 36% lower POA emissions from
anthropogenic combustion sources and 33% higher POA emissions from open
biomass burning sources on average over the decade 2000-2010 compared to the
reference simulation. The assumed volatility distributions are the same as in the
reference simulation. The decadal average global emission flux of primary organic
emissions in this case is 29 Tg yr$^{-1}$ from anthropogenic combustion sources and 38 Tg
yr$^{-1}$ from open biomass burning sources (Table 1).

**3.2 Sensitivity to chemistry**
The photooxidation of SVOCs and IVOCs emitted from fuel combustion and
biomass-burning sources can lead to the formation of substantial SOA mass on a
global scale (Jathar et al., 2011; Tsimpidi et al., 2014). To evaluate the sensitivity of
the model to the parameters used to describe the aging process we have conducted
three sensitivity simulations described below.



**High reaction rate constant:** In this simulation we investigate the sensitivity of the results to the rate constant used for the gas-phase photooxidation of SVOCs and IVOCs with OH. We assume that the corresponding oxidation rate constant is twice that of the reference simulation and equal to $4 \times 10^{-11}$ cm$^3$ molecules$^{-1}$ s$^{-1}$. All other parameters remained the same as in the reference simulation (Table 1).

**Alternative aging scheme:** To quantify the sensitivity of the results to the aging scheme, we designed a sensitivity case in which the aging scheme of Robinson et al. (2007) is used (Figure 1b). Based on this implementation, we are using nine volatility bins (compared to 5 in the reference simulation) to distribute the primary emissions into LVOCs ($10^{-2}$ and $10^{-1}$ µg m$^{-3}$), SVOCs ($10^0$, $10^1$, and $10^2$ µg m$^{-3}$), and IVOCs ($10^3$, $10^4$, $10^5$, and $10^6$ µg m$^{-3}$). This model set up is based on the formulation proposed by (Shrivastava et al., 2008). The volatility distribution of anthropogenic combustion and open biomass burning emissions is shown in Figure 2d. The sum of these emission factors is the same as in the reference simulation (2.5 for fuel combustion and 1 for biomass burning). However, the relative importance of SVOC and IVOC to total OA emissions is changed compared to the reference simulation. In the sensitivity simulation the fraction of SVOCs to the total emissions is 20% for fOA and 60% for bbOA (Figure 2d), compared to 32% and 70%, respectively, in the reference simulation (Figure 2a). Furthermore, the saturation concentration of the organic vapors reacting with OH is reduced by a factor of 10 (instead of 100 in the reference simulation) with a rate constant of $4 \times 10^{-11}$ cm$^3$ molec$^{-1}$ (double the value used in the reference simulation) and a 7.5% increase in mass to account for one added oxygen (half the value used in the reference simulation). The formation of SOA from LVOCs is possible in this configuration (contrary to the reference simulation) due to the presence of two species in the LVOC volatility range ($C^* < 3.2 \times 10^{-1}$ µg m$^{-3}$). Overall, in this simulation, 46 surrogate organic aerosol species are used to track the source- and volatility-resolved OA components compared to 26 aerosol species in the reference simulation.

**Hybrid aging scheme:** The reference and alternative aging scheme simulations assume that the volatility of the organic vapor precursors is reduced by two and one orders of magnitude, respectively, after each oxidation step. However, photooxidation





reactions of IVOCs can create products with a volatility 1 to 4 orders of magnitude
lower (Kroll and Seinfeld, 2008). Furthermore, recent experiments indicate that the
reduction in volatility due to oxidation reactions changes as the organic molecules
become more oxygenated and fragmentation becomes important (Chacon-Madrid et
al., 2013). To investigate the effect of these assumptions on the predicted global SOA
burden, we have modified the OA chemistry mechanism to include a hybrid method
to calculate the SOA formation from the oxidation of IVOCs based on the approach of
(Jathar et al., 2012). The SVOC oxidation scheme remains the same as in the
reference. The hybrid scheme distributes the IVOC first generation oxidation products
over a range of volatilities, with larger reductions in volatility compared to the
reference simulation. The oxidation of each IVOC is assumed to result in the
formation of two condensable organic gases with four and six orders of magnitude
lower volatility and aerosol yields equal to 0.71 and 0.115, respectively (Jathar et al.,
2014) (Figure 1c). Then, the reference oxidation scheme is used for subsequent
oxidation of these products assuming a factor of 100 reduction in volatility with 15%
increase in mass. The photo-oxidation of SVOCs and IVOCs in the hybrid aging
scheme is described by the following reactions:

$\quad\quad\quad\quad SVOC_i + OH \rightarrow 1.15\ SOG\text{-}sv_{i-1}$ $\quad\quad\quad\quad$ (R1)
$\quad\quad\quad\quad SOG\text{-}sv_i + OH \rightarrow 1.15\ SOG\text{-}sv_{i-1}$ $\quad\quad\quad\quad$ (R2)
$\quad\quad\quad\quad SOG\text{-}sv_i \leftrightarrow SOA\text{-}sv_i$ $\quad\quad\quad\quad$ (R3)

$\quad\quad\quad\quad IVOC_i + OH \rightarrow 0.71\ SOG\text{-}iv_{i-2} + 0.115\ SOG\text{-}iv_{i-3}$ $\quad$ (R4)
$\quad\quad\quad\quad SOG\text{-}iv_i + OH \rightarrow 1.15\ SOG\text{-}iv_{i-1}$ $\quad\quad\quad\quad$ (R5)
$\quad\quad\quad\quad SOG\text{-}iv_i \leftrightarrow SOA\text{-}iv_i$ $\quad\quad\quad\quad$ (R6)

This representation is more consistent with SOA formation from VOCs and provides
in principle at least a more realistic representation of SOA formation from IVOCs.

**3.3 Sensitivities to scavenging**
$\quad$ The wet and dry removal the organic vapours from the atmosphere depends on
their ability to partition into water which is commonly expressed by their Henry's law



constant (*H*). Two sensitivity simulations where performed to investigate the effect of
this uncertain parameter.

**Low solubility:** To test the sensitivity of the results to the solubility of the SVOC and
IVOC vapors, we have conducted a simulation using a Henry's law constant two
orders of magnitude lower than the reference and equal to $10^3$ M atm$^{-1}$ for both
primary and secondary SVOCs/IVOCs.

**Variable solubility:** The photochemical aging of organic vapors results on average in
less volatile and more hydrophilic products (Jimenez et al., 2009). To quantify the
effect of this change on the model results we have conducted a sensitivity simulation
in which the fresh SVOCs and IVOCs are hydrophobic with $H = 10$ M atm$^{-1}$ and
become more hydrophilic after their photochemical oxidation with an $H = 10^5$ M
atm$^{-1}$.

**4 Reference simulation results and evaluation**

325        The predicted decadal average surface concentrations of total OA, POA, SOA-sv,

and SOA-iv for the reference simulation are shown in Figure 3. High POA
concentrations are predicted over regions affected by biomass burning (i.e., the
tropical and boreal forests) as well as over the industrialized regions of the Northern
Hemisphere where strong fossil and biofuel combustion sources are located (i.e.,
Eastern and Southern Asia, Central and Eastern Europe, Western and Eastern US).
Further downwind of the sources, the POA concentration decreases substantially due
to dilution and evaporation (Figure 3b). On the other hand, the predicted SOA-sv and
SOA-iv concentrations are high over a wide area downwind of the polluted urban
areas and the major rainforests (Figure 3c and 3dc) due to the transport of IVOCs and
SVOCs and their continued chemical transformations. Since IVOC emissions from
anthropogenic sources are assumed to be two times higher than SVOC emissions
(Figure 1a), predicted SOA-iv is higher than SOA-sv over populated areas (Figure 3c
and 3d). On the other hand, over the tropical rainforests, SOA-sv and SOA-iv
concentrations are similar due to the low fraction of IVOCs assumed for the open
biomass burning OA emissions. Overall, the reference simulation yields a
tropospheric OA burden of 1.98 Tg consisting of 12% POA, 18% SOA-sv, 32% SOA-



iv, and 38% SOA-v. More details about the reference case results can be found in
Tsimpidi et al. (2016).
A comprehensive AMS dataset from field campaigns performed in the Northern
Hemisphere during 2001-2010 (Tsimpidi et al., 2016) has been used to evaluate the
model performance for each simulation. The mean bias (MB), mean absolute gross
error (MAGE), normalized mean bias (NMB), normalized mean error (NME), and the
root mean square error (RMSE) are used to assess the model performance for POA
(versus AMS hydrocarbon-like aerosol (HOA); Table 2) and SOA (versus AMS
oxygenated organic aerosol (OOA); Table 3). Tsimpidi et al. (2016) have shown that,
as expected the model underestimates the concentrations of POA and SOA over urban
locations due to its coarse resolution and missing sources in the emission database
(e.g., cold vehicle start and wood burning emissions in winter). Therefore, urban
locations are excluded from our analysis in order to avoid misinterpretation of the
sensitivity results and their effects on OA model performance. A comprehensive
analysis of the model evaluation based on the reference scenario results can be found
in Tsimpidi et al. (2016) and will be used here as a reference for analysing the effect
of each sensitivity scenario on the performance of the model. EMAC reproduces POA
levels with very little bias (NMB= -3%; Table 2). On the other hand, OOA
concentrations are underpredicted (-31%; Table 3) indicating that the model may be
missing an important source or formation pathway of SOA especially in winter
(Tsimpidi et al., 2016) or may be removing the corresponding pollutants faster.

**5 Sensitivity to emission factors**
**5.1 Low volatility**
In the first sensitivity test, the IVOC emissions are set to zero and only semi-
volatile organic compounds are emitted. This is accompanied by an increase of SVOC
emissions from anthropogenic and open biomass burning sources by 100% and 40%,
respectively. This initial partitioning of the emissions favors the particulate phase,
resulting in an increase of POA compared to the reference scenario (Figure 4a). The
largest fPOA and bbPOA increases are predicted over Eastern China (4.3 µg m$^{-3}$) and
the Congo Basin (3.9 µg m$^{-3}$), respectively. The higher SVOC emissions in the
sensitivity simulation result in an increase of the simulated SOA-sv concentrations as
well (Figure 5a). However, since a large fraction of the emitted SVOCs remains in the





particle phase, the SOA-sv concentration increase is smaller than the corresponding
changes in POA. Relatively strong fSOA-sv and bbSOA-sv increases are found over
the Indo-Gangetic Plane (IGP) (0.4 μg m$^{-3}$) and the Congo Basin (1.3 μg m$^{-3}$),
respectively. The "low volatility" simulation does not predict any SOA-iv as it
assumes zero IVOC emissions. Therefore, SOA-iv concentrations are zero around the
globe, resulting in substantial decreases in areas where the reference simulation
predicts high SOA-iv levels (Figures 3dc and 6a).

382         The significant decrease of organic emissions from anthropogenic sources (Table

1) due to the lack of IVOC emissions results in an overall decrease of total OA
concentrations by up to 5 μg m$^{-3}$ over anthropogenically polluted regions (Figure 7a).
On the other hand, organic emissions from open biomass burning sources remain at
the same level as the reference simulation (Table 1), however, they are assumed to
have lower volatility. This results in an increase of total OA concentrations in the
sensitivity simulation by up to 2 μg m$^{-3}$ over the tropical and boreal forests. Overall,
the calculated tropospheric burden of POA in the sensitivity simulation increases by
around 50% due to the increase of the SVOC emissions (Table 2). For the same
reason, the tropospheric fSOA-sv and bbSOA-sv burdens increase by 14% and 39%,
respectively. Nevertheless, the absence of IVOC emissions, and thus the significant
decrease of anthropogenic organic compound emissions, results in a decrease of the
total OA tropospheric burden by 23%. This result emphasizes the importance of the
volatility distributions used in the simulation and the contribution of IVOC emissions
to SOA formation on a global scale.

397         The simulated POA in the reference model configuration is very close to the

average HOA concentrations derived from the AMS measurements (Table 3).
Therefore, assuming lower volatility of the organic emissions results in overprediction
(NMB=43%). However, the performance of the model is significantly improved
during winter (Figure 8) since POA concentrations during that season were
underpredicted (NMB=-37%; Tsimpidi et al., 2016). On the other hand, during spring
the overestimate of POA increases in the sensitivity simulation (NMB=86%)
compared to the reference (NMB=26%). For summer and autumn, the performance of
the model changes from a slight underestimation of POA in the reference (NMB=-
15%) to a slight overprediction in the sensitivity test (NMB=30%). The performance
of the model in reproducing the OOA concentrations worsens in this sensitivity



simulation (Table 4). OOA was underpredicted by the model reference simulation
(NMB=-31%), therefore, by neglecting SOA formation from IVOC emissions in the
sensitivity run results in an even larger OOA underestimation (NMB=-52%). The
performance of the model does not change significantly during winter (Figure 8) since
the simulated SOA formation during this season is low (Tsimpidi et al., 2016). The
highest change in model performance occurs during spring when SOA is predicted to
reach the annual maximum (Tsimpidi et al., 2016); the predicted underestimation of
OOA increases from 20% in the reference to 50% in the sensitivity simulation. These
results indicate that the omission of IVOCs as a source of SOA in atmospheric models
can result in a significant underestimation of OA concentrations, especially during
periods where formation of SOA is strong.

**5.2 High IVOCs**
In the second sensitivity simulation, the increased IVOC emissions result in an
increase of total organics by 60%, and 150% from anthropogenic and open biomass
burning sources, respectively (Table 1). These additional organic emissions are
distributed only in the intermediate volatility bins, therefore, their impact on the
simulated POA and SOA-sv levels is marginal (Figures 4b and 5b, respectively). POA
increases, up to 0.6 μg m$^{-3}$ over Eastern China, while SOA-sv decreases, up to 0.3 μg
m$^{-3}$ over the Congo Basin. This effect can be explained by the assumption that SOA-
sv and SOA-iv form a pseudo-ideal solution. As a result, the increased SOA-iv
concentrations calculated in the sensitivity simulation favor the partitioning of the
fresh SVOCs into the aerosol phase, forming additional POA. At the same time,
SVOCs decrease in the gas-phase and therefore the formation of SOA-sv is reduced in
the sensitivity simulation. As expected, the largest effect is found for SOA-iv (Figure
6b). The significant increase of IVOC emissions results in large changes of SOA-iv
over areas close to anthropogenic sources (up to 5.7 μg m$^{-3}$ over the IGP) and biomass
burning regions (up to 5.3 μg m$^{-3}$ over the Congo Basin). The increase of SOA-iv
dominates the effect on total OA concentrations that increase up to 6 μg m$^{-3}$ (Figure
7b). Overall, the predicted changes of the tropospheric burden of POA and SOA-sv
are small (Table 2). However, the tropospheric burdens of fSOA-iv and bbSOA-iv
increase by 88% and 115%, respectively, resulting in an increase of the total OA
burden by 38%.





The additional IVOC emissions assumed in this sensitivity test do not affect the
performance of the model for POA. On the other hand, these additional emissions
bring the predicted SOA concentrations closer to the measured OOA levels (Table 4;
Figure 8). The NMB improves from -31% in the reference simulation to -10%. With
the exception of winter, where the model still underpredicts OOA levels (MB=2.2 µg
m$^{-3}$, Figure 8), the performance of the model for SOA improves with seasonal NMB
ranging from -16% (during summer) to 11% (during spring); compared to -33% and
-20% for the reference model, respectively. The improved performance of the model
due to the increase of IVOC emissions supports the hypothesis that the IVOC
emissions may have been underestimated in previous modeling studies that assumed
IVOC/POA =1.5 (Ots et al., 2016).

**5.3 Alternative POA emissions**
The final emission sensitivity test is used to estimate the uncertainty introduced by
the choice of emission database. The inventories used in the sensitivity simulation
assume 36% lower fuel combustion OA emissions and 33% higher biomass burning
OA emissions compared to the reference simulation, while the total OA emissions are
only reduced by 9%. Since the volatility distribution of the emissions is identical to
the reference simulation, the fractional changes of the calculated POA, SOA-sv, SOA-
iv are also similar (Table 4). The tropospheric burden of fOA (the sum of fPOA,
fSOA-sv, and fSOA-iv) decreases by 34%. On the other hand, bbOA (the sum of
bbPOA, bbSOA-sv, and bbSOA-iv) increases by 11%. Overall, the total tropospheric
OA burden increases by only 4%. The changes in fOA and bbOA concentrations,
however, are not spatially uniform. Over Europe, fOA decreases everywhere, up to
3.3 µg m$^{-3}$, except in Paris where fOA increases by 0.24 µg m$^{-3}$. Over the US fOA
slightly increases (mostly over the northeast by up to 0.6 µg m$^{-3}$), while it decreases
over Mexico by as much as 1.7 µg m$^{-3}$. The largest fOA change is predicted over Asia
where fOA decreases significantly, up to 8.3 µg m$^{-3}$, mostly over East Asia and the
IGP. bbOA decreases over the boreal forests (up to 3.6 µg m$^{-3}$), while it increases
significantly over the Southeast Asian tropical forests by up to 14 µg m$^{-3}$. Over the
Amazon and Congo forests, bbOA concentrations change significantly (the bbOA
changes vary from -2.4 to 3.3 µg m$^{-3}$ in the Amazon, and from -5.3 to 7.8 µg m$^{-3}$ in
Congo) but the average bbOA concentration over both regions remains the same.



Overall, the fOA and bbOA emission changes lead to total OA increases over the
tropical and boreal forests and decreases over anthropogenic areas (Figure 7c).
The lower OA emissions used in the sensitivity simulation (especially over China
and Europe) result in a reduction of both total POA and SOA concentrations (Tables 2
and 3). Consequently, the model now underestimates POA with NMB=-25% and
SOA with NMB=-40%. These results suggest that the use of the CMIP5 RCP4.5
emission inventory in EMAC results in OA concentrations that agree more closely to
the measurements compared to the AEROCOM database. It also underscores the large
uncertainty associated with primary OA emissions.

**6 Sensitivity to aging reactions**
**6.1 Higher aging reaction rate**
In this sensitivity simulation, the photochemical reaction rate constant for SVOCs
and IVOCs has been doubled compared to the reference. This results in an increase of
SOA-sv and SOA-iv concentrations worldwide (Figures 5d and 6d). SOA-sv
increases, up to 0.65 $\mu$g m$^{-3}$, mostly over the tropics and the polluted regions of
Eastern China and the IGP (Figure 5d). The effect on SOA-iv concentrations is even
more significant since IVOCs undergo more oxidation steps before forming SOA than
SVOCs. SOA-iv increased by up to 2.4 $\mu$g m$^{-3}$ mostly over the IGP and Eastern China
(Figure 6d). The SOA-iv increase over the tropics is smaller (up to 0.8 $\mu$g m$^{-3}$) due to
the assumed low fraction of IVOCs in biomass burning emissions. Overall, the
tropospheric burdens of SOA-sv and SOA-iv both increase by 0.04 Tg (or 11% and
7%, respectively). POA is not expected to be affected directly by the change of the
reaction rate constant. However, the substantial reduction of gas-phase SVOCs (due to
their increased reactivity) results in the re-evaporation of POA to achieve equilibrium,
reducing its concentration (Figure 4d) mainly over the tropics (up to 0.21 $\mu$g m$^{-3}$).
This results in an overall decrease of the tropospheric POA burden by 8%. Following
the significant increase of both SOA-sv and SOA-iv, total OA increases worldwide by
up to 3 $\mu$g m$^{-3}$ (Figure 7d). Overall, the tropospheric burden of total OA increases by

503   4%.

The model performance for POA is not affected by the change of the reaction rate
constant (Table 2) since POA remains largely unchanged over the Northern
Hemisphere (Figure 4d). On the other hand, the performance of the model regarding



SOA is significantly improved (Table 3). The underestimation of SOA by the model
is reduced (NMB=-22%) compared to the reference (NMB=-31%). The best
performance is found during spring (NMB=-7%) when the calculated SOA is almost
unbiased. However, during winter, the model still severely underestimates SOA
(NMB=-77%), which indicates that the gas-phase oxidation of SVOCs and IVOCs
does not suffice to explain the underprediction of SOA in winter.

**6.2 Alternative aging scheme**
In this sensitivity simulation we used the chemical aging scheme of Robinson et al.
(2007) which is currently the most commonly used in VBS models. This aging
scheme is accompanied by changes in the number of volatility bins used and the
assigned emission factors, the oxidation rate constant, the volatility reductions after
each oxidation step, and the increase in mass due to added oxygen (as discussed in
Sect. 3.2). The changes in the number of volatility bins and the emission factors used
for the SVOCs (Figure 2d) result in reduced condensation of SVOCs into the
particulate phase during the initial partitioning and therefore to a significant decrease
of POA (Figure 4e). The decrease of POA is global and most prominent over Eastern
China (up to 9.3 μg m$^{-3}$). This reflects a significant change in the tropospheric burdens
of both fPOA and bbPOA by 65% and 38%, respectively.
Furthermore, the reduced fraction of SVOCs to total OA emissions (see Section
3.2) results in a worldwide decrease of SOA-sv (Figure 5e) and an increase of SOA-iv
(Figure 6e). SOA-sv decreases up to 1.8 μg m$^{-3}$ over the Congo Basin and the IGP.
Similar to POA, the tropospheric burden of fSOA-sv and bbSOA-sv decreases by
68% and 47%, respectively. On the other hand, the increase in SOA-iv, due to the
increase in the IVOC fraction of the emissions, is not as strong as the decrease of
SOA-sv (Table 4). This is due to the slower aging in the sensitivity simulation (Figure
1b), compared to the reference (Figure 1a), which limits the formation of SOA from
IVOCs. SOA-iv increases up to 0.9 μg m$^{-3}$ over the Congo Basin and the IGP, while it
locally decreases by 0.1 μg m$^{-3}$ over Beijing, for example. The tropospheric burden of
fSOA-iv and bbSOA-iv increases by 14% and 30%, respectively. Overall, the sum of
SOA-sv and SOA-iv decreases by 7% due to the slower aging in this sensitivity
simulation. Following the simultaneous decrease of both POA and SOA, total OA



decreases worldwide by up to 11 μg m$^{-3}$ (Figure 7e) and its tropospheric burden is
reduced by 0.2 Tg (or 10%).
The reduction of both modelled POA and SOA results in reduced agreement of the
model with AMS measurements. Especially for POA, the modeled concentrations
decrease by 67% in the sensitivity simulation, resulting in a significant
underprediction of AMS-HOA (NMB=-67%). Modelled SOA also decreases (by
10%) in the sensitivity simulation, which degrades the model agreement with AMS-
OOA measurements (NMB=-38%). This sensitivity test underscores the significance
of the volatility distribution of the organic emissions and the associated aging scheme.

**6.3 Hybrid aging scheme**
The final chemistry sensitivity simulation focuses on the photochemical aging of
IVOCs and assumptions regarding the first oxidation step. The approach used here is
similar to the oxidation of the traditional VOCs, in contrast with the reference where
the oxidation of IVOCs produces only one product with two orders of magnitude
reduced volatility. However, the stoichiometric coefficient used in the reference
(equal to 1.15) is higher than the aerosol yields used in the sensitivity simulation
(Section 3.2). This results in a reduction of SOA-iv concentrations by up to 2.2 μg m$^{-3}$
(Figure 6f). Since the chemical scheme for SVOCs is identical in both the reference
and the sensitivity simulations, no significant change is found in either SOA-sv or
POA (Figure 5f and 4f, respectively). The decrease of SOA-iv concentrations has a
marginal effect on the initial partitioning of SVOC emissions resulting in slightly less
POA and more SOA-sv (by up to 0.1 μg m$^{-3}$ in either case). Therefore, total OA
concentrations are reduced worldwide following the decrease of SOA-iv. Overall, the
tropospheric burden of SOA-iv decreases by 37% in the sensitivity simulation
resulting in a decrease of total OA by 13% (Table 4).
The simulated POA concentrations remain almost unchanged in the sensitivity
simulation; therefore, similar to the reference, the calculated POA is unbiased
compared to measurements (Table 2). On the other hand, the lower SOA-iv
concentrations calculated by the model in this sensitivity test aggravate the
underestimation of OOA by the model (NMB=-39%). The decrease of modelled
SOA-iv concentrations is larger during spring (13%) and the calculated NMB for
SOA deteriorates from -20% in the reference to -30% in the sensitivity simulation.





**7 Sensitivity to wet/dry removal of organic vapors**

**7.1 Reduced Henry's law constant**

In this sensitivity test we used a Henry's law constant that is two orders of magnitude lower than in the reference simulation (see Section 3.3) for the gas-phase SVOCs and IVOCs. This change decreases their removal rate, thus increasing their lifetime and the concentrations of both POA (due to the condensation of the fresh SVOCs) and SOA (due to the condensation of the chemically aged SVOCs and IVOCs). POA increases up to 0.7 μg m$^{-3}$ over Eastern China (Figure 4g) where POA concentrations are relatively high (Figure 3b), however, the increase of POA in the rest of the world is less than 0.2 μg m$^{-3}$ (Figure 4g). SOA-sv increases up to 0.2 μg m$^{-3}$ mostly over the Congo Basin and the IGP (Figure 5g). The most significant change is calculated for SOA-iv. SOA-iv is formed from gases (i.e., IVOCs) that need to go through more than two oxidation steps to be able to condense to the aerosol phase (in comparison to only one oxidation step for SVOCs). Therefore, by lowering the Henry's law constant of IVOCs we prolong the lifetime of SOA-iv precursors, and their ability to undergo multiple oxidation steps and produce aerosols. This results in a significant increase of SOA-iv by up to 1.2 μg m$^{-3}$ (Figure 6g). Total OA increases by up to 2 μg m$^{-3}$ due to the simultaneous increase of both POA and SOA (Figure 7g). Overall, the tropospheric burden of SOA-iv increases by 17% and of total OA by 8%. It is also worth noticing that the tropospheric burden of fOA (sum of fPOA, fSOA-sv, and fSOA-iv) increases by 18% compared to an increase of 5% of the bbOA (sum of bbPOA, bbSOA-sv, and bbSOA-iv). The above results emphasize the significance of the removal of organic vapors for the calculated OA concentrations, and corroborate the importance of constraining the Henry' law constants of SVOCs and more importantly of IVOCs.

The change of Henry's law constant of SVOCs does not affect the model performance for POA significantly. POA slightly increases (by 4%), eliminating the already low model bias (Table 2). The SOA increase (by 12%) in the sensitivity simulation (mainly due to the increased SOA-iv) results in reduced SOA underestimation (Table 2). In both POA and SOA cases the effect is more important during winter, when wet removal is most efficient, and lower during summer. POA increases during winter by 10% while during summer it remains unchanged. SOA



increases during winter by 26% and during summer by only 3%, with spring and
autumn in between (~12%). Despite the wintertime POA and SOA increase in this
sensitivity simulation, the model still underestimates POA (NMB=-31%) and SOA
(NMB=-78%) during this season (Figure 8).

**7.2 Different Henry's law constant for POA and SOA**
In the last sensitivity test we assume that the freshly emitted SVOCs and IVOCs
are hydrophobic (with the Henry's law constant $H$ being 4 orders of magnitude lower
than the reference) while after photochemical aging $H$ increases to match the value
used in the reference (see Section 3.3). POA increases up to 0.7 $\mu g\ m^{-3}$, mostly over
Eastern China and to a lesser degree over Eastern Europe and Russia (Figure 4h).
SOA-sv increases up to 0.2 $\mu g\ m^{-3}$, mostly over the tropical forests of Central Africa
and Southeastern Asia, as well as over Eastern China and the IGP (Figure 5h). SOA-
iv also increases by up to 1 $\mu g\ m^{-3}$ (Figure 6h) because fresh IVOCs are more
hydrophobic in the sensitivity simulation, therefore, the time available to react with
OH is extended, forming additional SOA-iv. Total OA concentrations increase by up
to 2 $\mu g\ m^{-3}$ over Eastern China (Figure 7h). The tropospheric burden of total OA
increases by 8% in this sensitivity test with the strongest increase coming from fSOA-
iv (21%).
Both the predicted POA and SOA increase in the sensitivity simulation by 6% and
12% respectively. This results in a small overprediction of POA (NMB=4%),
compared to a small underprediction in the reference (NMB=-3%). For SOA, NMB
improves in the sensitivity simulation (NMB=-23%) compared to the reference (-
31%). Similar to the previous sensitivity test (Section 7.1) the effect is more relevant
during winter (POA and SOA increase by 9% and 36%, respectively), followed by
spring (POA and SOA increase by 8% and 16%, respectively) and autumn (POA and
SOA increase by 7% and 10%, respectively), and is small during summer (POA and
SOA increase by 2% and 5%, respectively) (Figures 8). This results in an improved
model performance for both POA and SOA during all seasons. The highest
improvement is found for SOA during spring when the NMB is reduced to -6% from
-20% in the reference. Despite the significant increase of SOA concentrations during
winter (by 36%), the model still strongly underestimates SOA (NMB=-76%),
indicating that the model underprediction of OOA cannot be attributed solely to





errors in the simulation of removal processes. Therefore, we expect that the
discrepancy in this season is related to sources that are missing or underestimated in
emission inventories, such as residential wood combustion in winter (Denier van der
Gon et al., 2015) and additional oxidation pathways.

**8 Summary and conclusions**
We investigated the effect of parameters and assumptions that control the
emissions, photochemical aging, and scavenging efficiency of LVOCs, SVOCs and
IVOCs on the simulated OA concentrations. We used the organic aerosol module
ORACLE, based on the VBS framework, in the EMAC global chemistry-climate
model. A global dataset of AMS measurements has been used to evaluate the
predicted POA and SOA concentrations, based on a number of sensitivity tests.
The results show that total OA concentrations are sensitive to the emissions of
IVOCs. By neglecting these emissions, the model produces unrealistically low SOA
concentrations resulting in the poorest model performance (NMB=-52%) compared to
the other eight simulations conducted (Table 3). Conversely, increasing the IVOC
emissions substantially improved the SOA model results, leading to the best model
performance (NMB=-10%). These results emphasize the need to accurately estimate
the IVOC emissions independently. The use of a more accurate POA emission
inventory is found to be of prime importance for the model performance, especially to
improve simulated POA concentrations in winter. In our tests, using an alternative
POA emission inventory led to a NMB of -25% compared to a low bias in the
performance of the reference model.
Sensitivity tests of the photochemical aging of SVOCs and IVOCs indicate the
importance of the OH-reaction rate. Assuming an increased reactivity of SVOC and
IVOC with OH improves the model results for SOA (NMB=-22%). This is even more
important for the IVOCs, which participate in a larger number of photochemical
reactions during atmospheric transport compared to the SVOCs. Another assumption
tested is that oxidation reactions of IVOCs are similar to many other VOCs, and
produce partly oxidized compounds with several orders of magnitude lower
volatilities. Despite the strong volatility reduction of the IVOC oxidation products, the
performance of the model was similar to the reference simulation since the IVOC
aerosol yields were lower compared to the stoichiometric coefficient used in the



reference. The use of an alternative aging scheme (based on Robinson et al., 2007)
resulted in lower SOA concentrations since the photochemical aging of SVOCs and
IVOCs was less effective. This led to a slight reduction in model performance for
SOA (Table 3). In this sensitivity test the fraction of SVOCs to total OA emissions
was lower compared to the reference, resulting in a significant reduction of POA and
a reduced model performance (NMB=-67%). This underscores the significance of the
assumed volatility distribution of OA emissions.
The calculated OA concentrations are highly sensitive to the scavenging
efficiency of the gas-phase SVOCs and IVOCs, expressed by the Henry's law
constant ($H$). Reducing $H$ resulted in an increase of both POA and SOA
concentrations, especially from the oxidation of IVOCs. This increase yielded
improved model performance, particularly for SOA (Table 3). Assuming different
hygroscopicity for the freshly emitted and the photochemically processed SVOCs and
IVOCs resulted in similar improvement of the model results (Tables 2 and 3). In this
sensitivity test, the simulated POA improved substantially during winter (NMB=-
29%) during which the model has difficulties reproducing AMS observations
(Tsimpidi et al., 2016). Nevertheless, SOA was still underpredicted during winter
(NMB=-76%) indicating that other processes (e.g., seasonally dependent emissions
and alternative oxidation paths) are a main cause of the inadequate performance.
Our results indicate that IVOCs can be major contributors to OA formation on a
global scale. However, their abundance and physicochemical properties are poorly
known, and more research is needed to determine the parameters that control their
emissions, chemistry, and atmospheric removal. According to the model results, a
combination of increased IVOC emissions, enhanced photochemical aging of IVOCs,
and decreased hygroscopicity of the freshly emitted IVOCs can help reduce
discrepancies between simulated SOA and observed OOA concentrations.

**9. Acknowledgements**

A.P. Tsimpidi acknowledges support from a DFG individual grand programme
(project reference TS 335/2-1) and V.A. Karydis acknowledges support from a FP7
Marie Curie Career Integration Grant (project reference 618349).




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






**Table 1.** Parameters used in the sensitivity simulations


| Simulation | Emission factor | | Emission rate (Tg yr$^{-1}$) | | Volatility bins | Reduction in volatility (µg m$^{-3}$) | Stoichiometric coefficient of aging reactions | Oxidation rate constant (cm$^3$ molec$^{-1}$ s$^{-1}$) | Henry's law constant (mol L$^{-1}$atm$^{-1}$) | |
|---|---|---|---|---|---|---|---|---|---|---|
| | fPOA | bbPOA | fPOA | bbPOA | | | | | Freshly emitted | Aged |
| Reference | 2.5 | 1 | 44.2 | 28.4 | 5 | $10^2$ | 1.15 | $2\times10^{-11}$ | $10^5$ | $10^5$ |
| Low volatility | 1 | 1 | 17.7 | 28.4 | 5 | $10^2$ | 1.15 | $2\times10^{-11}$ | $10^5$ | $10^5$ |
| High IVOCs | 4 | 2.5 | 70.7 | 71 | 5 | $10^2$ | 1.15 | $2\times10^{-11}$ | $10^5$ | $10^5$ |
| Alternative POA emissions | 2.5 | 1 | 28.5 | 37.8 | 5 | $10^2$ | 1.15 | $2\times10^{-11}$ | $10^5$ | $10^5$ |
| High reaction rate constant | 2.5 | 1 | 44.2 | 28.4 | 5 | $10^2$ | 1.15 | $4\times10^{-11}$ | $10^5$ | $10^5$ |
| Alternative aging scheme | 2.5 | 1 | 44.2 | 28.4 | 9 | 10 | 1.075 | $4\times10^{-11}$ | $10^5$ | $10^5$ |
| Hybrid aging scheme | 2.5 | 1 | 44.2 | 28.4 | 5 | SVOCs:$10^2$ IVOCs:$10^4$-$10^6$ | SVOCs:1.15 IVOCs:1.115-0.71 | SVOCs:$2\times10^{-11}$ IVOCs:$1.2\times10^{-11}$ | $10^5$ | $10^5$ |
| Low solubility | 2.5 | 1 | 44.2 | 28.4 | 5 | $10^2$ | 1.15 | $2\times10^{-11}$ | $10^3$ | $10^3$ |
| Variable solubility | 2.5 | 1 | 44.2 | 28.4 | 5 | $10^2$ | 1.15 | $2\times10^{-11}$ | 10 | $10^5$ |



**Table 2.** Statistical evaluation of EMAC POA (sum of fPOA and bbPOA) against
AMS POA (sum of HOA and BBOA) using 61 data sets in urban downwind and rural
areas during 2001-2010.

| Simulation Name | Mean Observed ($\mu g\ m^{-3}$) | Mean Predicted ($\mu g\ m^{-3}$) | MAGE ($\mu g\ m^{-3}$) | MB ($\mu g\ m^{-3}$) | NME (%) | NMB (%) | RMSE ($\mu g\ m^{-3}$) |
|---|---|---|---|---|---|---|---|
| Reference | | 0.51 | 0.38 | -0.02 | 71 | -3 | 0.50 |
| Low volatility | | 0.75 | 0.46 | 0.22 | 88 | 43 | 0.64 |
| High IVOCs | | 0.52 | 0.38 | -0.01 | 73 | 0 | 0.51 |
| Alternative POA emissions | | 0.39 | 0.33 | -0.14 | 63 | -25 | 0.44 |
| High reaction rate constant | 0.53 | 0.50 | 0.37 | -0.03 | 70 | -5 | 0.49 |
| Conservative aging scheme | | 0.17 | 0.42 | -0.36 | 79 | -67 | 0.60 |
| Hybrid aging scheme | | 0.50 | 0.38 | -0.03 | 72 | -4 | 0.50 |
| Low solubility | | 0.53 | 0.38 | 0 | 72 | 1 | 0.50 |
| Variable solubility | | 0.54 | 0.38 | 0.01 | 73 | 4 | 0.51 |





**Table 3.** Statistical evaluation of EMAC SOA against AMS OOA using 61 data sets
in downwind urban and rural areas during 2001-2010.

| Simulation Name | Mean Observed ($\mu g\ m^{-3}$) | Mean Predicted ($\mu g\ m^{-3}$) | MAGE ($\mu g\ m^{-3}$) | MB ($\mu g\ m^{-3}$) | NME (%) | NMB (%) | RMSE ($\mu g\ m^{-3}$) |
|---|---|---|---|---|---|---|---|
| Reference | | 1.91 | 1.39 | -0.87 | 50 | -31 | 2.02 |
| Low volatility | | 1.32 | 1.69 | -1.46 | 61 | -52 | 2.30 |
| High IVOCs | | 2.50 | 1.47 | -0.28 | 53 | -10 | 2.05 |
| Alternative POA emissions | | 1.66 | 1.55 | -1.12 | 56 | -40 | 2.15 |
| High reaction rate constant | 2.78 | 2.16 | 1.32 | -0.62 | 48 | -22 | 1.97 |
| Conservative aging scheme | | 1.73 | 1.49 | -1.05 | 53 | -38 | 2.09 |
| Hybrid aging scheme | | 1.71 | 1.46 | -1.08 | 53 | -39 | 2.08 |
| Low solubility | | 2.10 | 1.33 | -0.68 | 48 | -25 | 1.98 |
| Variable solubility | | 2.14 | 1.32 | -0.64 | 48 | -23 | 1.97 |







**Table 4.** Percentage change of the tropospheric burden of organic aerosol components for each sensitivity simulation relative to the reference simulation during the decade 2001-2010. Positive change corresponds to an increase. The predicted tropospheric burden in Tg of the reference simulation is also shown.

| | fPOA | bbPOA | fSOA-sv | bbSOA-sv | fSOA-iv | bbSOA-iv | Total OA |
|---|---|---|---|---|---|---|---|
| **Tropospheric burden of reference (Tg)** | 0.06 | 0.18 | 0.13 | 0.21 | 0.44 | 0.2 | 1.98 |
| | Percentage Change (%) from reference | | | | | | |
| **Simulation Name** | | | | | | | |
| Low volatility | 53 | 48 | 14 | 39 | -100 | -100 | -23 |
| High IVOCs | 7 | 5 | -3 | -4 | 88 | 165 | 38 |
| Alternative POA emissions | -39 | 10 | -33 | 11 | -34 | 11 | -8 |
| High reaction rate constant | -10 | -7 | 11 | 11 | 8 | 6 | 4 |
| Alternative aging scheme | -65 | -38 | -68 | -47 | 14 | 30 | -10 |
| Hybrid aging scheme | -2 | -1 | 2 | 2 | -37 | -36 | -13 |
| Low solubility | 6 | 1 | 11 | 4 | 21 | 8 | 8 |
| Variable solubility | 9 | 2 | 14 | 5 | 22 | 7 | 8 |












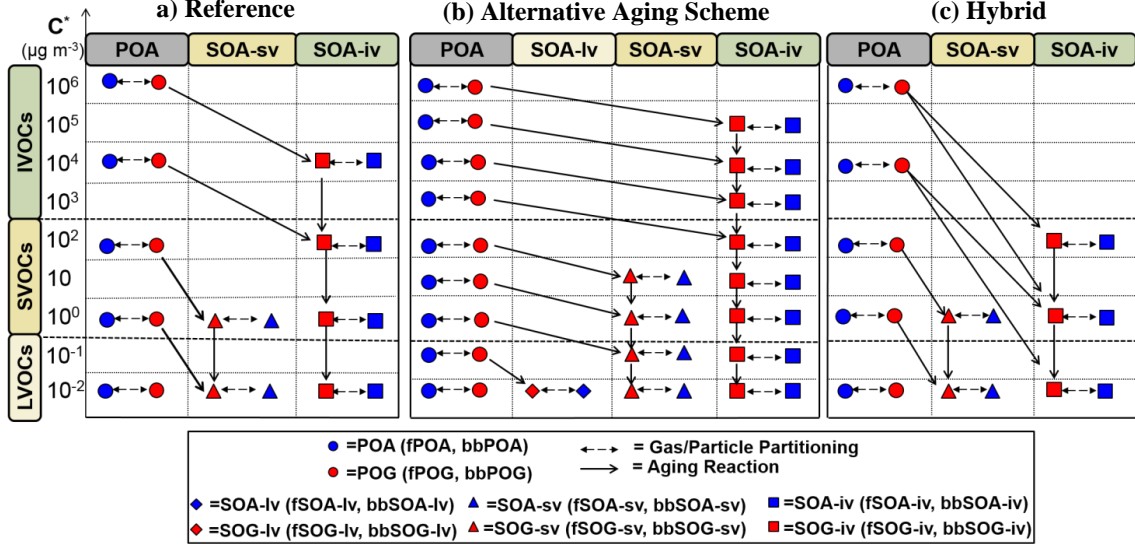

**Figure 1:** Schematic of the VBS resolution and the formation of SOA from SVOCs and IVOCs in the: (a) reference simulation, (b) alternative aging scheme and (c) hybrid case. SOA from LVOCs (SOA-lv) is only formed in the alternative aging scheme (b). Red indicates that the organic compound is in the vapor phase and blue in the particulate phase. The circles correspond to primary organics emitted as gases or particles. Diamonds symbolize the formation of SOA from LVOC emissions by fuel combustion and biomass burning. Triangles indicate SOA formation from SVOC emissions by fuel combustion and biomass burning, while the squares show SOA from IVOC by the same sources. Gas-aerosol partitioning, aging reactions, and names of species are also shown.





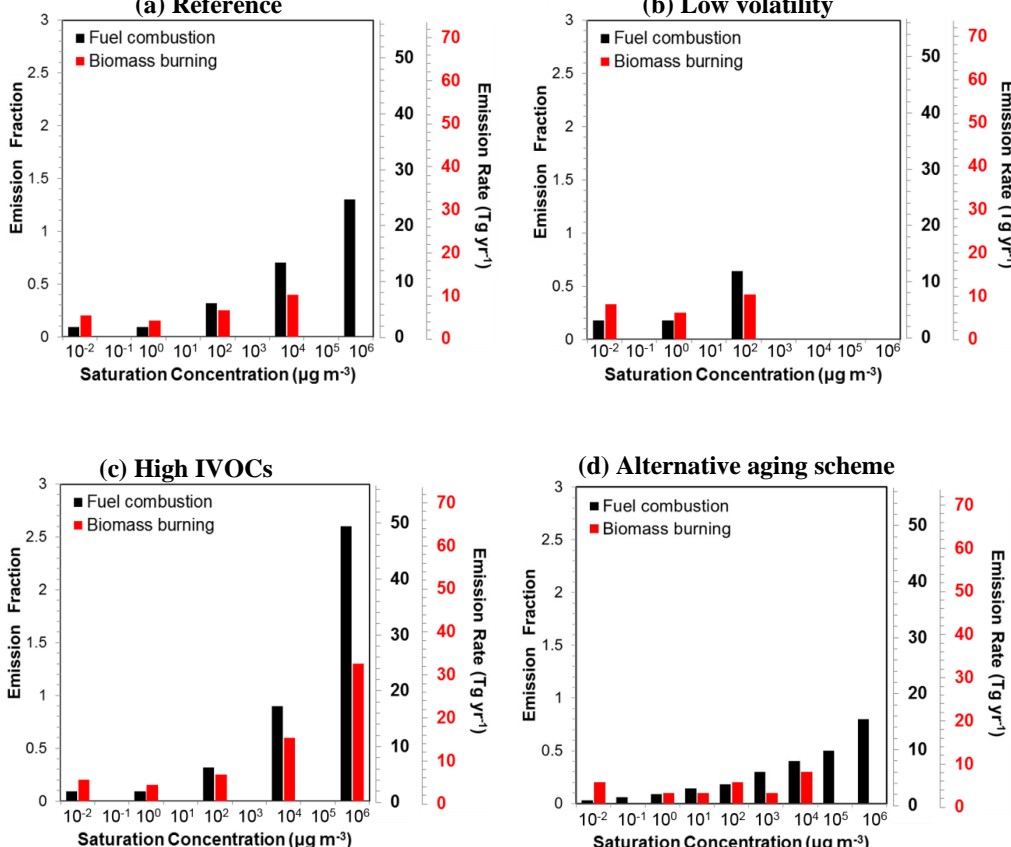

**Figure 2.** Volatility distribution for fuel combustion (black) and biomass burning OA (red) for the (a) reference, (b) low volatility, (c) high IVOCs and (d) conservative aging scheme simulations. The reference emission factors are from Robinson et al. (2007) for fPOA and May et al. (2013) for bbPOA emissions. The emission rates of fPOA and bbPOA are also shown on the right axis.





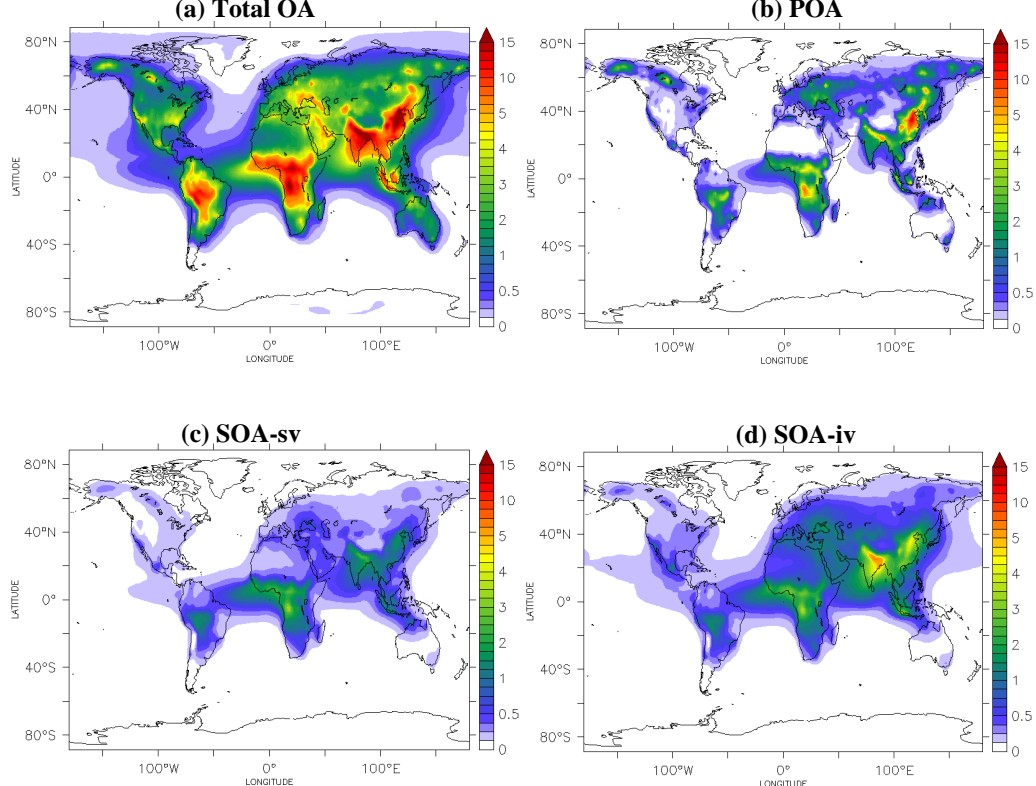

**Figure 3:** Predicted average surface concentrations (in μg m$^{-3}$) of: **(a)** Total OA (sum of POA, SOA-sv, SOA-iv and SOA-v), **(b)** POA and **(c)** SOA from the oxidation of SVOCs (SOA-sv) and **(d)** SOA from the oxidation of IVOCs (SOA-iv) for the reference simulation during the 2001-2010 period.





1000

**Figure 4:** Absolute changes (in µg m$^{-3}$) of the average surface POA concentrations between the reference and the **(a)** low volatility, **(b)** high IVOCs, **(c)** alternative POA emissions, **(d)** high reaction rate constant, **(e)** conservative aging scheme, **(f)** hybrid aging scheme, **(g)** low solubility, and, **(h)** hybrid solubility simulations during the period 2001-2010. A positive change indicates an increase in the sensitivity test.






**Figure 5:** Absolute changes (in µg m⁻³) of the average surface SOA concentrations from SVOCs (SOA-sv) between the reference and the **(a)** low volatility, **(b)** high IVOCs, **(c)** alternative POA emissions, **(d)** high reaction rate constant, **(e)** conservative aging scheme, **(f)** hybrid aging scheme, **(g)** low solubility, and, **(h)** hybrid solubility simulations during the period 2001-2010. A positive change indicates an increase in the sensitivity test.





**Figure 6:** Absolute changes (in μg m⁻³) of the average surface SOA concentrations from IVOCs (SOA-iv) between the reference and the **(a)** low volatility, **(b)** high IVOCs, **(c)** alternative POA emissions, **(d)** high reaction rate constant, **(e)** conservative aging scheme, **(f)** hybrid aging scheme, **(g)** low solubility, and, **(h)** hybrid solubility simulations during the period 2001-2010. A positive change indicates an increase in the sensitivity test.



**Figure 7:** Absolute changes (in μg m⁻³) of the average surface total OA concentrations between the reference and the **(a)** low volatility, **(b)** high IVOCs, **(c)** alternative POA emissions, **(d)** high reaction rate constant, **(e)** conservative aging scheme, **(f)** hybrid aging scheme, **(g)** low solubility, and, **(h)** hybrid solubility simulations during the period 2001-2010. A positive change indicates an increase in the sensitivity test.





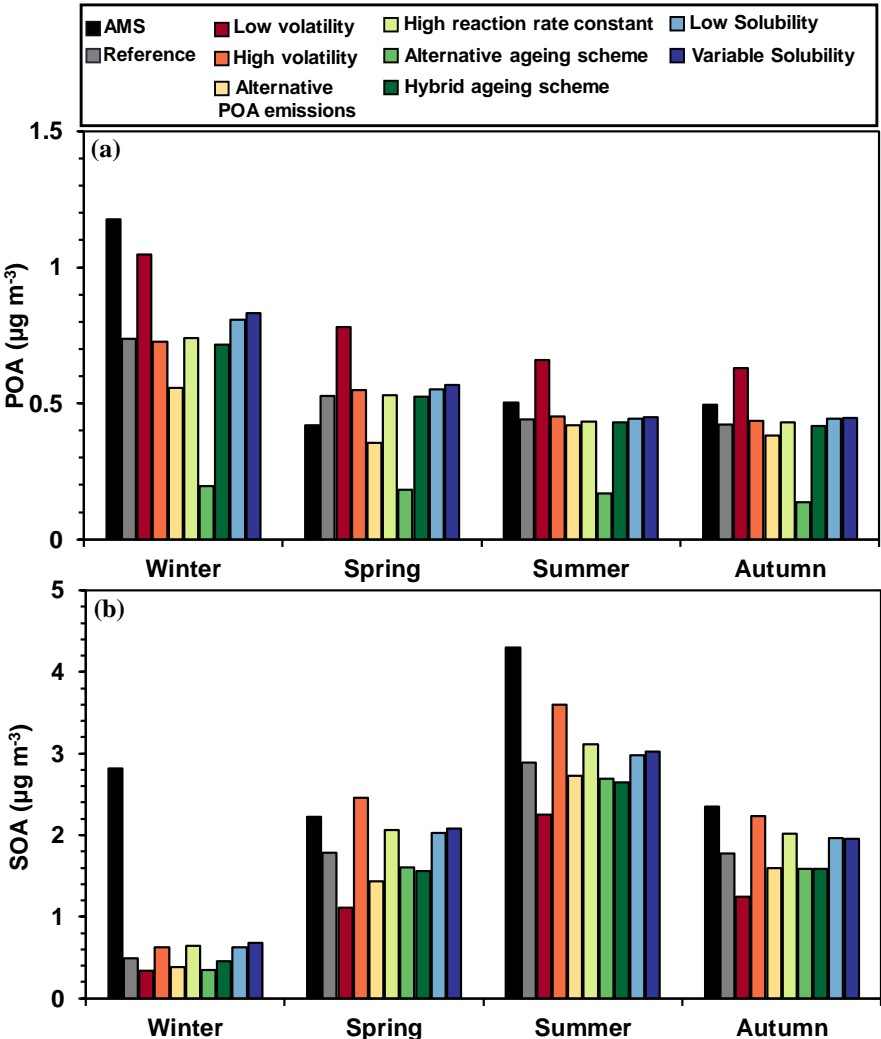

**Figure 8:** Average (a) POA and (b) SOA concentrations (in µg m⁻³) measured and predicted in the reference and sensitivity simulations during winter, spring, summer, and autumn in urban-downwind and rural areas of the continental Northern Hemisphere.