# Peer review of "Global-scale combustion sources of organic aerosols: Sensitivity to formation and removal mechanisms"

_Atmospheric Chemistry and Physics, 2017_

## Referee Comment (RC1) · Anonymous Referee #1 · 7 Mar 2017

Tsimpidi et al. quantify the global-scale contributions of combustion emissions to organic aerosols using a global model. Rather than a single value, the authors provide a range utilizing various inputs and parameters reported in the literature for modeling organic aerosols. Those sensitivities include variation in emissions (volatility of emissions, high estimates of IVOCs, an alternative POA emission inventory) alternative OA aging schemes, and alternative OA solubility parameters. The authors then compare results from the various sensitivity simulations against AMS measurements at rural locations.

The paper is generally well written and the analysis robust. I recommend the paper for publication but first would like to see a few clarifications and additional points listed

below discussed.

General Comments:

In recognizing there is a computational expense in a more explicit parameterization, are there benefits to utilizing different chemistry/aging schemes for anthropogenic and biogenic OA (e.g. Koo et al., 2014)?

Specific Comments:

There appears to be some inconsistency as to how the authors define IVOCs. On line 153, IVOCs are defined as having a C* between $10^4$ and $10^6$ $\mu$g m$^{-3}$. But on line 188-190, when discussing biomass burning emissions, the authors state:

"Biomass burning emissions are assumed to cover a range of volatilities from $10^{-2}$ to $10^4$ (May et al., 2013a), therefore, no IVOC emissions are assumed from biomass burning sources. . ."

Then, in the low volatility simulations, emissions of IVOCs are assumed to be zero. However, biogenic emissions in the reference simulation, which includes $10^4$ emissions, and the low volatility simulation are identical (28.4 Tg yr$^{-1}$).

In the low volatility simulations, how are the emissions from the $10^4$ bin that are not considered IVOCs redistributed to the lower bins? e.g. Total biogenic emissions are identical in the reference and low volatility simulations.

Line 219 and 220: The wording here makes it sound as if only emissions in the $10^4$ and $10^6$ bins are being increased by a factor of 1.5. Instead, I would recommend rewording this sentence to provide clarity. For example "increased by an additional factor of 1.5 times the POA emissions and then distributed in the volatility bins. . .". Also, how are they distributed, equally in the $10^4$ and $10^6$ bins? I'd also suggest making it more clear the total emissions in this case, that total anthropogenic emissions are 4x the POA inventory (1x L/SVOCs and 3x IVOCs) and biogenic emissions are 2.5x the

POA inventory (1x L/SVOCs and 1.5x IVOCs).

What is the reasoning to perform a model simulation with added IVOC emissions ($C^*$ of $10^6$) from biomass burning if measurements only support emissions up to a $C^*$ of $10^4$?

Line 448-451: Underestimates of IVOCs could be one cause of underpredictions, but could it also be other factors like uncertainty in yields (e.g. wall loss) or other missing precursors and/or pathways?
* * *

---

## Referee Comment (RC2) · Anonymous Referee #2 · 23 Mar 2017

In this paper, Tsimpidi et al. performed different sensitivity tests with the global chemistry-climate model EMAC in order to investigate the main parameters affecting the evolution of organic aerosol from combustion sources. Different assumptions on primary organic aerosol emission inventories, volatility distributions and reaction rate constants of SVOCs and IVOCs against OH are investigated. In addition, the authors deployed alternative aging schemes as well as different values of the Henry's law constant to test the effect of wet removal of SVOCs and IVOCs from the atmosphere. The ORACLE module, based on the VBS framework, is used within EMAC to model the evolution of OA in the atmosphere and results from the sensitivity tests compared against a comprehensive set of AMS measurements performed during 2001-2010.

The paper deserves publication, the results are well presented and the adopted schemes are appropriate for the analysis.

I recommend the paper for publication after considering the minor comments below.

Line 33: a more recent reference is needed.

Line 38: "*which can reduce their volatility*".
In recognizing that the main point of the sentence is to describe the formation of SOA, it would be desirable to mention the increase in volatility due to fragmentation as well.

Line 48: Please consider adding Jo et al., 2013 who has also investigated the effects of chemical aging on global secondary organic aerosol using the GEOS-Chem model and compared the model results against AMS datasets.

Line 142: What is the thickness of the first layer ? Please add this information.

Line 163-166: "*The volatilities of SVOCs and IVOCs are reduced by a factor of $10^2$ as a result of the OH reaction with a rate constant of $2x10^{-11}$ $cm^3$ $molecule^{-1}$ $s^{-1}$ and a 15% increase in mass to account for two added oxygens (Tsimpidi et al., 2014)*".
Does the model include any fragmentation pathways as well ? Please specify if fragmentation is directly/indirectly accounted for.

Line 170: Were shipping emissions taken into account ?

Line 359-362: "*On the other hand, OOA concentrations are underpredicted (-31%; Table 3) indicating that the model may be missing an important source or formation pathway of SOA especially in winter (Tsimpidi et al., 2016) or may be removing the corresponding pollutants faster*".
Please add the uncertainties in SOA yields due to wall loss in chambers as another possible reason for the underprediction of SOA. In the authors opinion, how much do vapor wall losses influence their results ?

Figures 4-5-6 and 7: In general, it seems that for all the sensitivity tests almost no changes are observed in the Scandinavian region. Is this simply because of the low SOA concentration predicted in this area of the domain ? Or are there other reasons ?

Line 638-641: "*Therefore, we expect that the discrepancy in this season is related to sources that are missing or underestimated in emission inventories, such as residential wood combustion in winter (Denier van der Gon et., 2015) and additional oxidation pathways*"
Here the important sources are clearly stated (i.e. residential wood combustion). Please add also explicitly the additional oxidation pathways that could be missing and the uncertainties in SOA yields due to wall loss in chambers.

Line 687-689: "*Nevertheless, SOA was still underpredicted during winter (NMB = -76%) indicating*

*that other processes (e.g., seasonally dependent emissions and alternative oxidation paths) are a main cause of the inadequate performance*"

Also in the conclusion part, I would explicitly mention the possible underestimation of residential wood combustion emissions as a possible reason for the underprediction of SOA during winter. Please consider adding more explicitly which additional oxidation pathways could be missing and again the uncertainties in SOA yields due to wall loss in chambers.

REFERENCE
Jo, D. S., Park, R. J., Kim, M. J. and Spracklen, D. V.: Effects of chemical aging on global secondary organic aerosol using the volatility basis set approach, Atmos. Environ., 81, 230–244, doi:10.1016/j.atmosenv.2013.08.055, 2013.

---

## Author Comment (AC1) · 2 May 2017

*Tsimpidi et al. quantify the global-scale contributions of combustion emissions to organic aerosols using a global model. Rather than a single value, the authors provide a range utilizing various inputs and parameters reported in the literature for modeling organic aerosols. Those sensitivities include variation in emissions (volatility of emissions, high estimates of IVOCs, an alternative POA emission inventory) alternative OA aging schemes, and alternative OA solubility parameters. The authors then compare results from the various sensitivity simulations against AMS measurements at rural locations.*

*The paper is generally well written and the analysis robust. I recommend the paper for publication but first would like to see a few clarifications and additional points listed below discussed.*

We would like to thank the reviewer for his/her positive response. Please see below our point by point response to reviewer's comments.

*General Comments:*

1. *In recognizing there is a computational expense in a more explicit parameterization, are there benefits to utilizing different chemistry/aging schemes for anthropogenic and biogenic OA (e.g. Koo et al., 2014)?*

According to previous modelling studies (Lane et al., 2008; Murphy and Pandis, 2009; Tsimpidi et al., 2014) aging of biogenic SOA may lead to significant over-predictions of OA over rural areas and forests. These findings are confirmed by observational studies that suggest that the aging of biogenic SOA does not result in a large change in its mass concentration (Ng et al., 2006; Donahue et al., 2012). Murphy et al. (2012) attributed this to a balancing of fragmentation and functionalization effects during the photochemical aging of biogenic SOA. On the other hand, the multigenerational chemistry of anthropogenic SOA precursors leads to a net average decrease of their volatility and increase of SOA production (Hildebrandt et al., 2009) and is often parameterized by regional and global models (Koo et al., 2014; Tsimpidi et al., 2016). Therefore, utilizing different chemistry/aging schemes for anthropogenic and biogenic OA gives us the opportunity to account for their different response in photochemical aging. This is now discussed in Section 2.2.

*Specific Comments:*

1. *There appears to be some inconsistency as to how the authors define IVOCs. On line 153, IVOCs are defined as having a $C^*$ between $10^4$ and $10^6$ $\mu g$ $m^{-3}$. But on line 188-190, when discussing biomass burning emissions, the authors state: "Biomass burning emissions are assumed to cover a range of volatilities from $10^{-2}$ to $10^4$ (May et al., 2013a), therefore, no IVOC emissions are assumed from biomass burning sources..." Then, in the low volatility simulations, emissions of IVOCs are assumed to be zero. However, biogenic emissions in the reference simulation, which includes $10^4$ emissions, and the low volatility simulation are identical (28.4 Tg $yr^{-1}$).*

We are sorry for this misunderstanding. The $10^4$ $\mu g$ $m^{-3}$ volatility bin represents IVOCs. Based on the findings of May et al. (2013) for biomass burning emissions, no additional IVOCs were included in these simulations. In the revised manuscript this specific sentence is rephrased as follows: "Biomass burning emissions are assumed to cover a range of volatilities from $10^{-2}$ to $10^4$ (May et al., 2013a) and no additional IVOC emissions are assumed from biomass burning sources. Therefore, the sum of their emission factors is unity (Figure 2a)". Furthermore, in the low volatility simulation, the sum of the emission factors is kept equal to unity by distributing the IVOC emissions (with $C^* = 10^4$ $\mu g$ $m^{-3}$) to lower volatility bins. Therefore, the biomass burning emission load is identical to the base case simulation, but distributed in lower volatility bins.

2. *In the low volatility simulations, how are the emissions from the $10^4$ bin that are not considered IVOCs redistributed to the lower bins? e.g. Total biogenic emissions are identical in the reference and low volatility simulations.*

The $10^4$ µg m$^{-3}$ volatility bin represents IVOC emissions. In the low volatility case, the 0.3 emission factor that was applied in the $10^4$ µg m$^{-3}$ volatility bin of the biomass burning emissions in the base case simulation is equally distributed to the $10^{-2}$, $10^0$ and $10^2$ µg m$^{-3}$ volatility bins by applying an extra 0.1 emission factor in each of these bins. Therefore, the total emission factor for the biomass burning emissions in both scenarios remained unchanged. This information is provided in Figure 2.

3. *Line 219 and 220: The wording here makes it sound as if only emissions in the $10^4$ and $10^6$ bins are being increased by a factor of 1.5. Instead, I would recommend rewording this sentence to provide clarity. For example "increased by an additional factor of 1.5 times the POA emissions and then distributed in the volatility bins...". Also, how are they distributed, equally in the $10^4$ and $10^6$ bins? I'd also suggest making it more clear the total emissions in this case, that total anthropogenic emissions are 4x the POA inventory (1x L/SVOCs and 3x IVOCs) and biogenic emissions are 2.5x the POA inventory (1x L/SVOCs and 1.5x IVOCs).*

Following the reviewer's recommendation, we have rephrased these lines in the revised manuscript as follows: "To estimate an upper limit of the IVOC contribution to the formation of SOA, a sensitivity simulation is conducted in which the emissions of IVOCs are increased by 1.5 times the original POA emissions. These extra emissions are distributed in the volatility bins with $C^*$ of $10^4$ and $10^6$ µg m$^{-3}$ (Figure 2c) by applying an additional emission factor of 0.5 and 1 respectively. The LVOC and SVOC emissions are the same as in the reference simulation. Overall, the total anthropogenic and biomass burning emissions are 4 and 2.5 times higher respectively than the original POA emission inventory. The decadal average global emission flux of primary organic emissions in this sensitivity test is 71 Tg yr$^{-1}$ for both anthropogenic and open biomass burning sources (Table 1)".

4. *What is the reasoning to perform a model simulation with added IVOC emissions ($C^*$ of $10^6$) from biomass burning if measurements only support emissions up to a $C^*$ of $10^4$?*

The May et al. (2013) volatility distribution for biomass burning OA, used in this work, are derived from thermodenuder measurements covering a range of volatilities with $C^*$ from $10^{-2}$ to $10^4$ µg m$^{-3}$. However, this range can be extended to even higher volatilities. Agaki et al. (2011) estimated that the unspeciated nonmethane organic compound (NMOC) emissions account for 50% of the total observed NMOC. Jathar et al. (2014) reported that 20% of the NMOC emissions are not speciated and are currently misclassified in emission inventories. Given that the unspeciated organic emissions are still largely uncertain, we performed a sensitivity simulation by adding organic emissions outside the range of May et al. (2013)'s volatility distribution (i.e., for $C^*$ equal to $10^6$ µg m$^{-3}$).

5. *Line 448-451: Underestimates of IVOCs could be one cause of underpredictions, but could it also be other factors like uncertainty in yields (e.g. wall loss) or other missing precursors and/or pathways?*

That is correct. In this study, we have conducted multiple sensitivity scenarios in order to quantify the impact of different parameters on the predicted OA concentration. It is worth mentioning that in all cases tested the model underestimates OA. This suggests that the source of undreprediction of OA by atmospheric chemistry models reported in the literature (Tsigaridis et al., 2014) cannot be attributed to only one cause rather to a combination of different factors. Potential causes that

are not explored here (e.g., uncertainties in SOA yields due to wall losses in chambers, missing sources and oxidation pathways, etc.) can also play a role, especially during the winter period. This is already discussed in the conclusions of our manuscript but is now more emphasized in the revised manuscript.

**References**

Akagi, S. K., Yokelson, R. J., Wiedinmyer, C., Alvarado, M. J., Reid, J. S., Karl, T., Crounse, J. D., and Wennberg, P. O.: Emission factors for open and domestic biomass burning for use in atmospheric models, Atmos. Chem. Phys., 11, 4039-4072, 2011.

Donahue, N. M., Henry, K. M., Mentel, T. F., Kiendler-Scharr, A., Spindler, C., Bohn, B., Brauers, T., Dorn, H. P., Fuchs, H., Tillmann, R., Wahner, A., Saathoff, H., Naumann, K.-H., Moehler, O., Leisner, T., Mueller, L., Reinnig, M.-C., Hoffmann, T., Salo, K., Hallquist, M., Frosch, M., Bilde, M., Tritscher, T., Barmet, P., Praplan, A. P., DeCarlo, P. F., Dommen, J., Prevot, A. S. H., and Baltensperger, U.: Aging of biogenic secondary organic aerosol via gas-phase OH radical reactions, Proceedings of the National Academy of Sciences of the United States of America, 109, 13503-13508, 2012.

Hildebrandt, L., Donahue, N. M., and Pandis, S. N.: High formation of secondary organic aerosol from the photo-oxidation of toluene, Atmospheric Chemistry and Physics, 9, 2973-2986, 2009.

Jathar, S. H., Gordon, T. D., Hennigan, C. J., Pye, H. O. T., Pouliot, G., Adams, P. J., Donahue, N. M., and Robinson, A. L.: Unspeciated organic emissions from combustion sources and their influence on the secondary organic aerosol budget in the United States, Proceedings of the National Academy of Sciences of the United States of America, 111, 10473-10478, 2014.

Koo, B., Knipping, E., and Yarwood, G.: 1.5-Dimensional volatility basis set approach for modeling organic aerosol in CAMx and CMAQ, Atmospheric Environment, 95, 158-164, 2014.

Lane, T. E., Donahue, N. M., and Pandis, S. N.: Simulating secondary organic aerosol formation using the volatility basis-set approach in a chemical transport model, Atmos. Environ., 42, 7439-7451, 2008.

May, A. A., Levin, E. J. T., Hennigan, C. J., Riipinen, I., Lee, T., Collett, J. L., Jimenez, J. L., Kreidenweis, S. M., and Robinson, A. L.: Gas-particle partitioning of primary organic aerosol emissions: 3. Biomass burning, Journal of Geophysical Research-Atmospheres, 118, 11327-11338, 2013.

Murphy, B. N., Donahue, N. M., Fountoukis, C., Dall'Osto, M., O'Dowd, C., Kiendler-Scharr, A., and Pandis, S. N.: Functionalization and fragmentation during ambient organic aerosol aging: application of the 2-D volatility basis set to field studies, Atmospheric Chemistry and Physics, 12, 10797-10816, 2012.

Murphy, B. N. and Pandis, S. N.: Simulating the formation of semivolatile primary and secondary organic aerosol in a regional chemical transport model, Environ. Sci. Technol., 43, 4722-4728, 2009.

Ng, N. L., Kroll, J. H., Keywood, M. D., Bahreini, R., Varutbangkul, V., Flagan, R. C., Seinfeld, J. H., Lee, A., and Goldstein, A. H.: Contribution of first- versus second-generation products to secondary organic aerosols formed in the oxidation of biogenic hydrocarbons, Environ. Sci. Technol., 40, 2283-2297, 2006.

Tsigaridis, K., Daskalakis, N., Kanakidou, M., Adams, P. J., Artaxo, P., Bahadur, R., Balkanski, Y., Bauer, S. E., Bellouin, N., Benedetti, A., Bergman, T., Berntsen, T. K., Beukes, J. P., Bian, H., Carslaw, K. S., Chin, M., Curci, G., Diehl, T., Easter, R. C., Ghan, S. J., Gong, S. L., Hodzic, A., Hoyle, C. R., Iversen, T., Jathar, S., Jimenez, J. L., Kaiser, J. W., Kirkevag, A., Koch, D., Kokkola, H., Lee, Y. H., Lin, G., Liu, X., Luo, G., Ma, X., Mann, G. W., Mihalopoulos, N., Morcrette, J. J., Mueller, J. F., Myhre, G., Myriokefalitakis, S., Ng, N. L., O'Donnell, D., Penner, J. E., Pozzoli, L., Pringle, K. J., Russell, L. M., Schulz, M., Sciare, J., Seland, O., Shindell, D. T., Sillman, S., Skeie, R. B., Spracklen, D., Stavrakou, T., Steenrod, S. D., Takemura, T., Tiitta, P., Tilmes, S., Tost, H., van Noije, T., van Zyl, P. G., von Salzen, K., Yu, F., Wang, Z., Wang, Z., Zaveri, R. A., Zhang, H., Zhang, K., Zhang, Q., and Zhang, X.: The AeroCom evaluation and intercomparison of organic aerosol in global models, Atmospheric Chemistry and Physics, 14, 10845-10895, 2014.

Tsimpidi, A. P., Karydis, V. A., Pandis, S. N., and Lelieveld, J.: Global combustion sources of organic aerosols: model comparison with 84 AMS factor-analysis data sets, Atmos. Chem. Phys., 16, 8939-8962, 2016.

Tsimpidi, A. P., Karydis, V. A., Pozzer, A., Pandis, S. N., and Lelieveld, J.: ORACLE (v1.0): module to simulate the organic aerosol composition and evolution in the atmosphere, Geoscientific Model Development, 7, 3153-3172, 2014.

---

## Author Comment (AC2) · 2 May 2017

*In this paper, Tsimpidi et al. performed different sensitivity tests with the global chemistry-climate model EMAC in order to investigate the main parameters affecting the evolution of organic aerosol from combustion sources. Different assumptions on primary organic aerosol emission inventories, volatility distributions and reaction rate constants of SVOCs and IVOCs against OH are investigated. In addition, the authors deployed alternative aging schemes as well as different values of the Henry's law constant to test the effect of wet removal of SVOCs and IVOCs from the atmosphere. The ORACLE module, based on the VBS framework, is used within EMAC to model the evolution of OA in the atmosphere and results from the sensitivity tests compared against a comprehensive set of AMS measurements performed during 2001-2010.*

*The paper deserves publication, the results are well presented and the adopted schemes are appropriate for the analysis.*

*I recommend the paper for publication after considering the minor comments below.*

We would like to thank the reviewer for his/her positive response. Below is our point by point response to the reviewer's comments.

1. *Line 33: a more recent reference is needed.*

   The sentence has been changed to "Organic aerosol (OA) is an important constituent of the atmosphere, contributing about 50% of the total submicron dry aerosol mass (Zhang et al., 2011) with major impacts on human health and climate (IPCC, 2013; Lelieveld et al., 2015)."

2. *Line 38: "which can reduce their volatility". In recognizing that the main point of the sentence is to describe the formation of SOA, it would be desirable to mention the increase in volatility due to fragmentation as well.*

   Following the suggestion of the reviewer we have rewritten the sentence as follows: "The co-emitted organic vapors can undergo one or more chemical transformations, which can alter their volatility due to functionalization (reducing their volatility) or fragmentation (increasing their volatility). The oxidation products with lower volatility can be transferred to the particulate phase forming secondary organic aerosol (SOA)."

3. *Line 48: Please consider adding Jo et al., 2013 who has also investigated the effects of chemical aging on global secondary organic aerosol using the GEOS-Chem model and compared the model results against AMS datasets.*

   Done.

4. *Line 142: What is the thickness of the first layer? Please add this information.*

   It is 68 m. We have added this information in the revised manuscript.

5. *Line 163-166: "The volatilities of SVOCs and IVOCs are reduced by a factor of $10^2$ as a result of the OH reaction with a rate constant of $2 \times 10^{-11}$ $cm^3$ $molecule^{-1}$ $s^{-1}$ and a 15% increase in mass to account for two added oxygens (Tsimpidi et al., 2014)". Does the model include any fragmentation pathways as well? Please specify if fragmentation is directly/indirectly accounted for.*

   The model does not include explicitly the fragmentation pathway. This has been indirectly taken into account by assuming that the functionalization and fragmentation processes result in a net average decrease of volatility for SOA produced by SVOC/IVOC and anthropogenic VOC and no

net average change of volatility for SOA produced by biogenic VOC (Murphy et al., 2012). This information has been added in the revised manuscript.

6. *Line 170: Were shipping emissions taken into account?*

   Yes. Shipping emissions are part of the CMIP5 RCP4.5 emission inventory.

7. *Line 359-362: "On the other hand, OOA concentrations are underpredicted (-31%; Table 3) indicating that the model may be missing an important source or formation pathway of SOA especially in winter (Tsimpidi et al., 2016) or may be removing the corresponding pollutants faster". Please add the uncertainties in SOA yields due to wall loss in chambers as another possible reason for the underprediction of SOA. In the authors opinion, how much do vapor wall losses influence their results?*

   Thank you for the helpful suggestion. The loss of semi-volatile vapors to the walls of laboratory chambers has been added as a possible reason for the underprediction of OOA. According to Zhang et al. (2014), these vapor losses can lead to substantially underestimated SOA formation.

8. *Figures 4-5-6 and 7: In general, it seems that for all the sensitivity tests almost no changes are observed in the Scandinavian region. Is this simply because of the low SOA concentration predicted in this area of the domain? Or are there other reasons?*

   Figures 4-7 depict absolute changes of OA concentrations for each sensitivity test; therefore, changes are low due to the low OA concentrations predicted by the model over the Scandinavian region (Figure 3) in the base case scenario. Changes are only noticeable for SOA-iv (Figure 6), which is the dominant OA component in the southern Scandinavian region (Figure 3).

9. *Line 638-641: "Therefore, we expect that the discrepancy in this season is related to sources that are missing or underestimated in emission inventories, such as residential wood combustion in winter (Denier van der Gon et., 2015) and additional oxidation pathways" Here the important sources are clearly stated (i.e. residential wood combustion). Please add also explicitly the additional oxidation pathways that could be missing and the uncertainties in SOA yields due to wall loss in chambers.*

   Aqueous-phase and heterogeneous oxidation reactions of organics are not included in our model and can be considered as a possible cause of the OOA underestimation. This information, together with the wall losses in chambers as an additional source of uncertainty, has been added in the revised text.

10. *Line 687-689: "Nevertheless, SOA was still underpredicted during winter (NMB = -76%) indicating that other processes (e.g., seasonally dependent emissions and alternative oxidation paths) are a main cause of the inadequate performance" Also in the conclusion part, I would explicitly mention the possible underestimation of residential wood combustion emissions as a possible reason for the underprediction of SOA during winter. Please consider adding more explicitly which additional oxidation pathways could be missing and again the uncertainties in SOA yields due to wall loss in chambers.*

    We have revised the text accordingly.

**References**

IPCC: (Intergovernmental Panel on Climate Change): The physical science basis. Contribution of working group I to the fifth assessment report of the intergovernmental panel on climate change. T.F. Stocker, D. Qin, G.-K. Plattner, M. Tignor, S.K. Allen, J. Boschung, A. Nauels, Y. Xia, V.

Bex, and P.M. Midgley (eds.). Cambridge University Press, Cambridge, United Kingdom and New York, NY, USA, 2013. 2013.

Jo, D. S., Park, R. J., Kim, M. J., and Spracklen, D. V.: Effects of chemical aging on global secondary organic aerosol using the volatility basis set approach, Atmos. Environ., 81, 230-244, 2013.

Lelieveld, J., Evans, J. S., Fnais, M., Giannadaki, D., and Pozzer, A.: The contribution of outdoor air pollution sources to premature mortality on a global scale, Nature, 525, 367-371, 2015.

Murphy, B. N., Donahue, N. M., Fountoukis, C., Dall'Osto, M., O'Dowd, C., Kiendler-Scharr, A., and Pandis, S. N.: Functionalization and fragmentation during ambient organic aerosol aging: application of the 2-D volatility basis set to field studies, Atmospheric Chemistry and Physics, 12, 10797-10816, 2012.

Zhang, Q., Jimenez, J. L., Canagaratna, M. R., Ulbrich, I. M., Ng, N. L., Worsnop, D. R., and Sun, Y. L.: Understanding atmospheric organic aerosols via factor analysis of aerosol mass spectrometry: a review, Anal. Bioanal. Chem., 401, 3045-3067, 2011.

---

## Author Response (AR1)

Tsimpidi et al. quantify the global-scale contributions of combustion emissions to organic aerosols using a global model. Rather than a single value, the authors provide a range utilizing various inputs and parameters reported in the literature for modeling organic aerosols. Those sensitivities include variation in emissions (volatility of emissions, high estimates of IVOCs, an alternative POA emission inventory) alternative OA aging schemes, and alternative OA solubility parameters. The authors then compare results from the various sensitivity simulations against AMS measurements at rural locations.

The paper is generally well written and the analysis robust. I recommend the paper for publication but first would like to see a few clarifications and additional points listed below discussed.

We would like to thank the reviewer for his/her positive response. Please see below our point by point response to reviewer's comments.

**General Comments:**

1. In recognizing there is a computational expense in a more explicit parameterization, are there benefits to utilizing different chemistry/aging schemes for anthropogenic and biogenic OA (e.g. Koo et al., 2014)?

According to previous modelling studies (Lane et al., 2008; Murphy and Pandis, 2009; Tsimpidi et al., 2014) aging of biogenic SOA may lead to significant overpredictions of OA over rural areas and forests. These findings are confirmed by observational studies that suggest that the aging of biogenic SOA does not result in a large change in its mass concentration (Ng et al., 2006; Donahue et al., 2012). Murphy et al. (2012) attributed this to a balancing of fragmentation and functionalization effects during the photochemical aging of biogenic SOA. On the other hand, the multigenerational chemistry of anthropogenic SOA production (Hildebrandt et al., 2009) and is often parameterized by regional and global models (Koo et al., 2014; Tsimpidi et al., 2016). Therefore, utilizing different chemistry/aging schemes for anthropogenic and biogenic OA gives us the opportunity to account for their different response in photochemical aging. This is now discussed in Section 2.2.

**Specific Comments:**

 There appears to be some inconsistency as to how the authors define IVOCs. On line 153, IVOCs are defined as having a C\* between 104 and 106 μg m-3. But on line 188-190, when discussing biomass burning emissions, the authors state: "Biomass burning emissions are assumed to cover a range of volatilities from 10-2 to 104 (May et al., 2013a), therefore, no IVOC emissions are assumed from biomass burning sources..." Then, in the low volatility simulations, emissions of IVOCs are assumed to be zero. However, biogenic emissions in the reference simulation, which includes  $10^4$  emissions, and the low volatility simulation are identical (28.4 Tg yr-1).

We are sorry for this misunderstanding. The  $10^4 \ \mu g \ m^{-3}$  volatility bin represents IVOCs. Based on the findings of May et al. (2013) for biomass burning emissions, no additional IVOCs were included in these simulations. In the revised manuscript this specific sentence is rephrased as follows: "Biomass burning emissions are assumed to cover a range of volatilities from  $10^{-2}$  to  $10^4$  (May et al., 2013a) and no additional IVOC emissions are assumed from biomass burning sources. Therefore, the sum of their emission factors is unity (Figure 2a)". Furthermore, in the low volatility simulation, the sum of the emission factors is kept equal to unity by distributing the IVOC emissions (with C\* =  $10^4 \ \mu g \ m^{-3}$ ) to lower volatility bins. Therefore, the biomass burning emission load is identical to the base case simulation, but distributed in lower volatility bins.

2. In the low volatility simulations, how are the emissions from the 104 bin that are not considered IVOCs redistributed to the lower bins? e.g. Total biogenic emissions are identical in the reference and low volatility simulations.

The  $10^4 \,\mu\text{g} \text{ m}^{-3}$  volatility bin represents IVOC emissions. In the low volatility case, the 0.3 emission factor that was applied in the  $10^4 \,\mu\text{g} \text{ m}^{-3}$  volatility bin of the biomass burning emissions in the base case simulation is equally distributed to the  $10^{-2}$ ,  $10^0$  and  $10^2 \,\mu\text{g} \text{ m}^{-3}$  volatility bins by applying an extra 0.1 emission factor in each of these bins. Therefore, the total emission factor for the biomass burning emissions in both scenarios remained unchanged. This information is provided in Figure 2.

3. Line 219 and 220: The wording here makes it sound as if only emissions in the 104 and 106 bins are being increased by a factor of 1.5. Instead, I would recommend rewording this sentence to provide clarity. For example "increased by an additional factor of 1.5 times the POA emissions and then distributed in the volatility bins…". Also, how are they distributed, equally in the 104 and 106 bins? I'd also suggest making it more clear the total emissions in this case, that total anthropogenic emissions are 4x the POA inventory (1x L/SVOCs and 3x IVOCs) and biogenic emissions are 2.5x the POA inventory (1x L/SVOCs and 1.5x IVOCs).

Following the reviewer's recommendation, we have rephrased these lines in the revised manuscript as follows: "To estimate an upper limit of the IVOC contribution to the formation of SOA, a sensitivity simulation is conducted in which the emissions of IVOCs are increased by 1.5 times the original POA emissions. These extra emissions are distributed in the volatility bins with C\* of

 $10^4$  and  $10^6$  µg m-3 (Figure 2c) by applying an additional emission factor of 0.5 and 1 respectively. The LVOC and SVOC emissions are the same as in the reference simulation. Overall, the total anthropogenic and biomass burning emissions are 4 and 2.5 times higher respectively than the original POA emission inventory. The decadal average global emission flux of primary organic emissions in this sensitivity test is 71 Tg yr-1 for both anthropogenic and open biomass burning sources (Table 1)".

4. What is the reasoning to perform a model simulation with added IVOC emissions (C\* of 106) from biomass burning if measurements only support emissions up to a C\* of 104?

The May et al. (2013) volatility distribution for biomass burning OA, used in this work, are derived from thermodenuder measurements covering a range of volatilities with  $C^*$  from  $10^{-2}$  to  $10^4 \,\mu \text{g m}^{-3}$ . However, this range can be extended to even higher volatilities. Agaki et al. (2011) estimated that the unspeciated nonmethane organic compound (NMOC) emissions account for 50% of the total observed NMOC. Jathar et al. (2014) reported that 20% of the NMOC emissions are not speciated and are currently misclassified in emission inventories. Given that the unspeciated organic emissions are still largely uncertain, we performed a sensitivity simulation by adding organic emissions outside the range of May et al. (2013)'s volatility distribution (i.e., for  $C^*$  equal to  $10^6 \,\mu \text{g m}^{-3}$ ).

5. Line 448-451: Underestimates of IVOCs could be one cause of underpredictions, but could it also be other factors like uncertainty in yields (e.g. wall loss) or other missing precursors and/or pathways?

That is correct. In this study, we have conducted multiple sensitivity scenarios in order to quantify the impact of different parameters on the predicted OA concentration. It is worth mentioning that in all cases tested the model underestimates OA. This suggests that the source of undreprediction of OA by atmospheric chemistry models reported in the literature (Tsigaridis et al., 2014) cannot be attributed to only one cause rather to a combination of different factors. Potential causes that are not explored here (e.g., uncertainties in SOA yields due to wall losses in chambers, missing sources and oxidation pathways, etc.) can also play a role, especially during the winter period. This is already discussed in the conclusions of our manuscript but is now more emphasized in the revised manuscript.

[revised manuscript text omitted]

969 in downwind urban and rural areas during 2001-2010.

| 9 | 7 | 0 |  |
|---|---|---|--|
|   |   |   |  |

|                             | Mean                  | Mean                  | MAGE                  | MB                    | NME | NMB | RMSE                  |
|-----------------------------|-----------------------|-----------------------|-----------------------|-----------------------|-----|-----|-----------------------|
| Simulation Name             | Observed              | Predicted             | (µg m -3 ) | (µg m -3 ) | (%) | (%) | (µg m -3 ) |
|                             | (µg m -3 ) | (µg m -3 ) |                       |                       |     |     |                       |
| Reference                   |                       | 1.91                  | 1.39                  | -0.87                 | 50  | -31 | 2.02                  |
| Low volatility              |                       | 1.32                  | 1.69                  | -1.46                 | 61  | -52 | 2.30                  |
| High IVOCs                  | 2.78                  | 2.50                  | 1.47                  | -0.28                 | 53  | -10 | 2.05                  |
| Alternative POA emissions   |                       | 1.66                  | 1.55                  | -1.12                 | 56  | -40 | 2.15                  |
| High reaction rate constant |                       | 2.16                  | 1.32                  | -0.62                 | 48  | -22 | 1.97                  |
| Conservative aging scheme   |                       | 1.73                  | 1.49                  | -1.05                 | 53  | -38 | 2.09                  |
| Hybrid aging scheme         |                       | 1.71                  | 1.46                  | -1.08                 | 53  | -39 | 2.08                  |
| Low solubility              |                       | 2.10                  | 1.33                  | -0.68                 | 48  | -25 | 1.98                  |
| Variable solubility         |                       | 2.14                  | 1.32                  | -0.64                 | 48  | -23 | 1.97                  |

973 Table 4. Percentage change of the tropospheric burden of organic aerosol components for

each sensitivity simulation relative to the reference simulation during the decade 2001-2010.Positive change corresponds to an increase. The predicted tropospheric burden in Tg of the

976 reference simulation is also shown.

|                                       | fPOA | bbPOA | fSOA-sv    | bbSOA-sv   | fSOA-iv     | bbSOA-iv | Total OA |
|---------------------------------------|------|-------|------------|------------|-------------|----------|----------|
| Tropospheric burden of reference (Tg) | 0.06 | 0.18  | 0.13       | 0.21       | 0.44        | 0.2      | 1.98     |
|                                       |      |       | Percentage | Change (%) | from refere | ence     |          |
| Simulation Name                       |      |       |            |            |             |          |          |
| Low volatility                        | 53   | 48    | 14         | 39         | -100        | -100     | -23      |
| High IVOCs                            | 7    | 5     | -3         | -4         | 88          | 165      | 38       |
| Alternative POA emissions             | -39  | 10    | -33        | 11         | -34         | 11       | -8       |
| High reaction rate constant           | -10  | -7    | 11         | 11         | 8           | 6        | 4        |
| Alternative aging scheme              | -65  | -38   | -68        | -47        | 14          | 30       | -10      |
| Hybrid aging scheme                   | -2   | -1    | 2          | 2          | -37         | -36      | -13      |
| Low solubility                        | 6    | 1     | 11         | 4          | 21          | 8        | 8        |
| Variable solubility                   | 9    | 2     | 14         | 5          | 22          | 7        | 8        |

| 977 |  |
|-----|--|
| 978 |  |
| 979 |  |
| 980 |  |
| 981 |  |
| 982 |  |
| 983 |  |